# NMDAR-dependent long-term depression is associated with increased short term plasticity through autophagy mediated loss of PSD-95

Benjamin Compans [1,6], Come Camus [1,6], Emmanouela Kallergi[2], Silvia Sposini[1], Magalie Martineau[1], Corey Butler [1], Adel Kechkar[1], Remco V. Klaassen[3], Natacha Retailleau[1], Terrence J. Sejnowski[4], August B. Smit [3], Jean-Baptiste Sibarita [1], Thomas M. Bartol Jr[4], David Perrais [1], Vassiliki Nikoletopoulou[2], Daniel Choquet [1,5,7] & Eric Hosy [1,7 ✉]

Long-term depression (LTD) of synaptic strength can take multiple forms and contribute to circuit remodeling, memory encoding or erasure. The generic term LTD encompasses various induction pathways, including activation of NMDA, mGlu or P2X receptors. However, the associated specific molecular mechanisms and effects on synaptic physiology are still unclear. We here compare how NMDAR- or P2XR-dependent LTD affect synaptic nanoscale organization and function in rodents. While both LTDs are associated with a loss and reorganization of synaptic AMPARs, only NMDAR-dependent LTD induction triggers a profound reorganization of PSD-95. This modification, which requires the autophagy machinery to remove the T19-phosphorylated form of PSD-95 from synapses, leads to an increase in AMPAR surface mobility. We demonstrate that these post-synaptic changes that occur specifically during NMDAR-dependent LTD result in an increased short-term plasticity improving neuronal responsiveness of depressed synapses. Our results establish that P2XR- and NMDAR-mediated LTD are associated to functionally distinct forms of LTD.

[1] Interdisciplinary Institute for Neuroscience, CNRS, Univ. Bordeaux, IINS, UMR 5297, Bordeaux, France. [2] Department of Fundamental Neurosciences, University of Lausanne, Lausanne, Switzerland. [3] Department Molecular and Cellular Neurobiology, Amsterdam, HV, The Netherlands. [4] Howard Hughes Medical Institute, Salk Institute for Biological Studies, La Jolla, CA, USA. [5] Univ. Bordeaux, CNRS, INSERM, Bordeaux Imaging Center, BIC, UMS 3420, Bordeaux, France. [6] These authors contributed equally: Benjamin Compans, Come Camus. [7] These authors jointly supervised this work: Daniel Choquet, Eric Hosy. ✉email: eric.hosy@u-bordeaux.fr

Changes in synaptic efficacy, either by strengthening through long-term potentiation (LTP) or weakening through long-term depression (LTD), are believed to underlie learning and memory. At the archetypal Schaeffer collateral to CA1 pyramidal neuron synapses, these forms of activity-dependent synaptic plasticity rely largely on the regulation of AMPA-type glutamate receptor number at synapses. In addition to endo- and exocytosis which control the total amount of receptors at the neuronal surface, the equilibrium at the membrane between freely-diffusive and immobilized AMPAR at the PSD regulates synaptic responses[1–6]. This equilibrium involves the binding of AMPAR associated-proteins to post-synaptic scaffolding proteins (e.g., PSD-95), which anchor AMPAR complexes at the PSD. A balanced activity of phosphatases and kinases on various targets ultimately regulates the affinities of the components of the AMPAR complexes for their scaffolds and their characteristic trapping times at synapses[7–9]. We, and others, have demonstrated that synaptic strength is not solely dependent on the quantity of glutamate per pre-synaptic vesicle and the number of post-synaptic AMPARs, but also rely on their nanoscale organization with respect to the pre-synaptic active zone[10–13]. Modeling predicts that long-term modifications in synaptic strength, as observed during LTD or LTP, can result from (i) changes in AMPAR density at synapse, (ii) variations in receptor amount per cluster or (iii) modifications of the alignment between pre-synaptic release site and AMPAR clusters[10,14].

This dynamic equilibrium between mobile and trapped AMPARs regulates synaptic transmission properties at multiple timescales. Indeed, at short time scales (ms to s), we established the role of mobile AMPAR in short-term plasticity. This plasticity, observed when synapses are stimulated at tens of Hz, was previously thought to rely on both pre-synaptic mechanisms, and AMPAR desensitization properties[15–17]. After glutamate release, desensitized AMPARs can rapidly exchange by lateral diffusion with naive receptors, increasing the number of receptors which can be activated after a second glutamate release[18–22]. In various experimental paradigms, a decrease in the pool of mobile AMPARs using either receptor crosslinking[18,19], CaMKII activation[20] or fused AMPAR-TARPs[18,23] enhances short-term depression. In addition to its role in short-term plasticity, AMPAR lateral diffusion and activity-dependent synaptic trapping plays a key role during LTP[20,24]. Specifically, altering the synaptic recruitment of AMPAR by interfering with AMPAR lateral diffusion impairs the early LTP[24]. These studies demonstrated that nanoscale regulation of AMPAR organization and dynamics tune individual as well as frequency dependent synaptic responses. However, few studies have yet addressed the molecular reshuffling induced by LTD.

LTD is a generic term indicating a decrease in synaptic strength, but it can be induced by different pathways. These include for example the classical glutamate-induced LTD through the activation of NMDAR or mGluR[25,26], the insulin-induced LTD[27] or the ATP-induced LTD[28,29]. Each of these forms results from a specific physiological stimulus such as low frequency stimulation (LFS), which mainly involves NMDAR[26], or the release of ATP by astrocytes following noradrenergic stimulation[29]. Interestingly, they all share intertwined molecular pathways, activating either specific or identical phosphatases/ kinases[27,29,30]. All these pathways lead to a rapid increase of AMPAR endocytosis, responsible for synaptic depression. However, it is still unclear if all these LTD types trigger at long term, similar modification in synaptic physiology.

To investigate how various LTD protocols are associated with the dynamic reorganization of synaptic AMPAR, we combined live and fixed super-resolution microscopy, live imaging of glutamate release and measurements of exo-endocytosis of AMPARs, electrophysiology together with modeling. We tested two well characterized LTD protocols based on the activation of either the NMDARs, by NMDA, or the P2X receptors (P2XRs), by ATP[26,29,31]. We observed that NMDAR-, but not P2XR-dependent LTD triggers specific changes in PSD-95 nanoscale organization in an autophagosome-dependent mechanism, responsible for an increase in AMPAR lateral diffusion. Finally, we show that the latter improves the capacity of depressed synapses to integrate high frequency stimulations. Overall, our results reveal that following the initial decrease in AMPAR synaptic content by endocytosis, the various LTD forms are not only associated to a depression of the post-synaptic response, but trigger specific modifications of the synaptic architecture and physiology affecting the ability of the post-synapse to integrate pre-synaptic inputs.

## Results

**ATP and NMDA treatments both induce a long-lasting decrease in synaptic AMPAR content and miniature amplitude.** We performed direct Stochastic Optical Reconstruction Microscopy (dSTORM) experiments and electrophysiological recordings in cultured rat primary hippocampal neurons to monitor AMPAR organization and currents following the application of either ATP or NMDA, two chemical protocols well-established to trigger a long lasting synaptic depression[29,31]. The fluorescent emission property of individual AMPAR was extracted from isolated receptors present at the membrane surface[12] and used to estimate the density and number of AMPARs in different neuronal compartments.

NMDA treatment (30 μM, 3 min) led to a rapid and stable decrease in AMPAR density both in the dendritic shaft and spines (decrease of 40% in dendrites and 22% in spines, Supplementary Fig. 1A, B). As previously described, ~50% of synaptic AMPAR are organized in nanodomains facing pre-synaptic release sites[11,12]. NMDAR-dependent LTD was also associated with a rapid (within the first 10 min following NMDA application) and stable (up to 3 h) depletion in AMPAR content per nanodomain (estimated number of AMPAR per nanodomain (mean of the mean per cell): t0: 19.77 ± 1.20, t10: 12.82 ± 1.05, t30: 12.59 ± 0.82) (Fig. 1A, B). In contrast, the overall nanodomain diameter was preserved as shown by the stability of their full width half maximum (t0: 79.13 ± 1.82 nm, t10: 82.04 ± 2.53 nm, t30: 84.35 ± 2.43 nm) (Fig. 1C). This reorganization of synaptic AMPAR was associated with a depression in AMPAR-mediated miniature excitatory post-synaptic current amplitude (mEPSC; Fig. 1D, E, t0: 11.34 ± 0.50 pA, t10: 7.60 ± 0.49 pA, t30: 8.00 ± 0.65 pA), which lasted up to 3 h after NMDA treatment (Fig. 1F, t0: 10.98 ± 0.73 pA, t180: 7.06 ± 0.49 pA).

In parallel, we measured the effect of ATP treatment, reported as able to induce a robust and long lasting LTD, on both AMPAR nanoscale organization and mEPSCs (Fig. 1G–L). Purinergic receptors from the P2X family were activated using ATP (100 μM, 1 min) in the presence of CGS15943 (3 μM) to avoid adenosine receptor activation[29,32]. As for NMDA, ATP treatment triggered a rapid and long-lasting decrease in AMPAR content both at dendritic shafts, spines (Supplementary Fig. 1E, F) and nanodomains (estimated number of AMPAR per nanodomain: t0: 20.36 ± 1.61, t10: 13.21 ± 0.46, t30: 15.38 ± 0.53), without affecting their overall organization (t0: 68.19 ± 2.55 nm, t10: 66.81 ± 2.57 nm, t30: 70.62 ± 3.11 nm) (Fig. 1G–I). In parallel, this depletion of AMPAR nanodomains was associated with a stable long-lasting decrease in mEPSC amplitude (t0: 11.93 ± 0.89 pA, t10: 9.05 ± 0.68 pA, t30: 8.95 ± 0.70 pA) (Fig. 1J–L).

Altogether, these results indicate that both NMDA- and ATP-induced synaptic depression are associated with a reduction in the number of surface AMPARs at the synapse and on the dendrite,

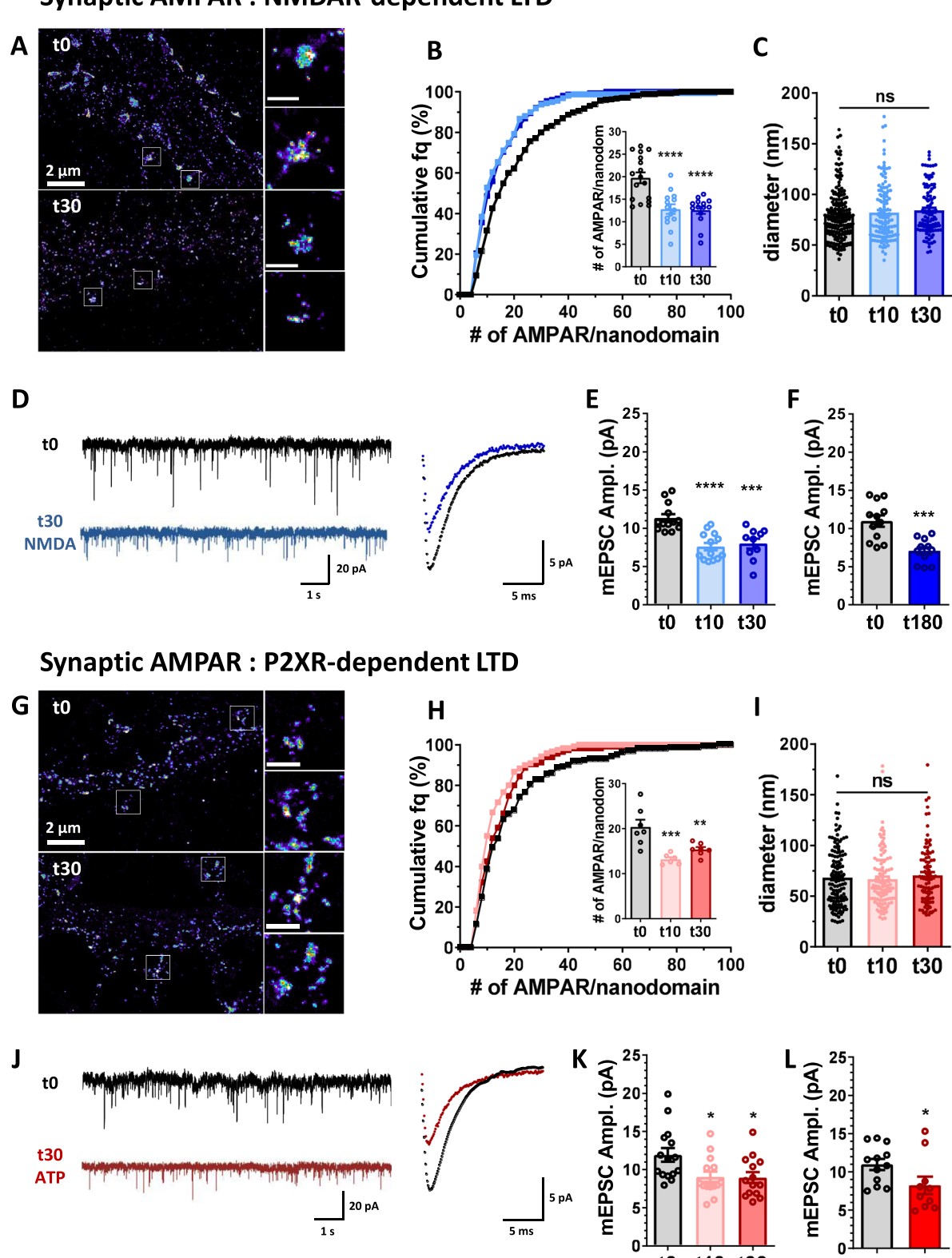

**Synaptic AMPAR : NMDAR-dependent LTD**

**Synaptic AMPAR : P2XR-dependent LTD**

notably leading to a depletion in nanodomain content, without a change in their overall dimensions.

To determine whether the mEPSC amplitude decrease was associated with modifications in current kinetics, which might be caused by changes in the composition of AMPAR complexes, we analyzed both rise and decay times of mEPSCs. We observed no

modifications in current kinetics when LTD was induced by either NMDA or ATP (Supplementary Fig. 2). Interestingly, we observed a transient decrease in the mEPSC frequency after NMDA treatment, which might be explained by the decrease in the number of nanodomains per spine (Supplementary Figs. 1C, D and 2). This observation suggests the complete disappearance of some domains.

**Fig. 1 NMDA and ATP application triggers a rapid and long-lasting nanoscale reorganization of AMPAR at synapses associated to a long-term synaptic current depression. A** Example of super-resolution intensity images of a piece of dendrite obtained using dSTORM technique on live stained neurons for endogenous GluA2 containing AMPARs at basal state (t0) or 30 min (t30) following NMDA application (30 μM, 3 min). Enlarged synapses are shown on the right. **B** Cumulative distribution of nanodomain AMPAR content ($n = 275$, 159 and 152 for t0, t10 and t30 respectively), and in the inset, the mean per cell. The number of AMPARs per nanodomain was estimated 0, 10 and 30 min following NMDA treatment as explained in Nair et al. 2013 (mean ± SEM, $n = 17$, 14 and 14 respectively, one-way ANOVA, $p < 0.0001$ and Dunnett's post-test found significant differences between t10 or t30 and t0, $p < 0.0001$). Nanodomain content is significantly decreased 10 and 30 min following NMDA treatment compared to non-treated cells. **C** Diameter of AMPAR synaptic nanodomains. Nanodomain sizes were measured by anisotropic Gaussian fitting of pre-segmented clusters obtained on dSTORM images. Nanodomain diameter (mean ± SEM) 0, 10 and 30 min following NMDA treatment are plotted ($n = 191$, 127 and 100 respectively, one-way ANOVA, $p = 0.2487$). Nanodomain size is not affected by NMDA application. **D** Left panel: example of miniature EPSC traces recorded on cultured neurons in basal condition (dark trace) or 30 min after NMDA treatment (blue trace). Right panel: Superposition of a mean trace of AMPAR mEPSC in basal (dark) and 30 min post-NMDA treatment (blue). **E** and **F** Average of the mESPC amplitude recorded on neurons 0, 10 or 30 min (**E**) and 180 min (**F**) after NMDA treatment. Miniature EPSC amplitudes are significantly depressed 10 and 30 min after NMDA treatment (**E**, $n = 13$, 13 and 10 respectively, one-way ANOVA $p < 0.0001$ and Dunnett's post-test found significant differences $p < 0.0001$ and $p = 0.0003$ between t0 and t10, and t0 and t30 respectively), and this depression stays for at least 3 h (**F**, $n = 12$ and 11 respectively, t-test $p = 0.0003$). **G** Example of super-resolution intensity images of a piece of dendrite obtained using dSTORM technique on neurons live stained for endogenous GluA2 containing AMPARs at basal state (t0) or 30 min (t30) following ATP (100 μM, 1 min). Enlarged synapses are shown on the right. **H** Cumulative distribution of nanodomain AMPAR content ($n = 158$, 120 and 115 for t0, t10 and t30 respectively), and in the inset, the mean per cell. The number of AMPARs per nanodomains was estimated 0, 10 and 30 min following ATP treatment (mean ± SEM, $n = 7$, 6 and 7 respectively, one-way ANOVA, $p = 0.0006$ and Dunnett's post-test found significant differences between t10 or t30 and t0, $p = 0.0004$ and $p = 0.0063$ respectively). Nanodomain content is decreased 10 and 30 min following ATP treatment compared to non-treated cells. **I** Measure of nanodomain diameter is not affected 0, 10 and 30 min following ATP treatment ($n = 130$, 112 and 91 respectively, one-way ANOVA, $p = 0.6391$). **J** Left panel: example of miniature EPSC traces recorded on cultured neurons in basal condition (dark trace) or 30 min after ATP treatment (red trace). Right panel: Superposition of a mean trace of AMPAR mEPSC in basal (dark) and 30 min post-ATP treatment (red). **K** and **L** Average of the mESPC amplitudes recorded on neurons 0, 10 or 30 min (**K**) and 180 min (**L**) after ATP treatment (100 μM, 1 min). Synaptic transmission (mEPSCs) is significantly depressed 10 and 30 min after ATP treatment (mean ± SEM, **K**, $n = 15$, 14 and 14 respectively, one-way ANOVA $p = 0.0124$ and Dunnett's post-test found significant differences $p = 0.0214$ and $p = 0.0172$ between t0 and t10, and t0 and t30 respectively), and this depression stays for at least 3 h (**L**, $n = 12$ and 10 respectively, t-test $p = 0.0485$). Scale bars (**A** and **G**) left images = 2 μm, zoom on synapses (left panels) = 500 nm.

This effect on the nanodomains was not observed when LTD was induced by ATP (Supplementary Figs. 1G and 2).

As demonstrated in[10], a decrease in synaptic response could be due to a change in pre- to post-synaptic alignment. To estimate whether NMDA treatment affected the trans-synaptic organization, we performed dual-color d-STORM experiments to measure the alignment of the pre-synaptic protein RIM1/2 with the post-synaptic AMPAR (Supplementary Fig. 3A, B). We calculated the centroid to centroid distances between RIM1/2 and GluA2-containing AMPAR clusters at t0, and 10 and 30 min after NMDA treatment (Supplementary Fig. 3A, D). NMDA treatment did not significantly change the RIM/AMPAR co-organization.

**AMPAR lateral diffusion is increased during NMDAR-dependent LTD but not during P2XR-dependent LTD.** Changes in both AMPARs endocytosis and exocytosis have been involved in synaptic plasticities. More particularly, in NMDA- and ATP-induced LTD, a rapid but transient increase in endocytosis rate has been observed[29,33–35]. This mechanism has been proposed to primarily mediate the depression of synaptic AMPAR currents. The general view being thus that induction of both NMDAR- and P2XR-dependent LTD results from transient AMPAR endocytosis at perisynaptic sites after their escape from synapses by lateral diffusion. However, both NMDAR- and P2XR-dependent forms of LTD last at least for 3 h (Fig. 1F and L). This indicates that once synapses are depressed, they reach a new equilibrium, which is poorly characterized.

We thus aimed to first determine the status of AMPAR trafficking at the equilibrium at depressed synapses, as a function of the LTD induction protocol. To this aim, we investigated the properties of AMPAR lateral diffusion during the two LTD-inducing protocols. Using the single-particle tracking technique uPAINT[36], we measured endogenous GluA2-containing AMPAR mobility at the cell surface. We acquired time lapse series consisting of 1.5 min acquisitions performed every 5 min for 30 min on cells in the basal condition and upon NMDAR- or P2XR-dependent LTD induction. As previously described[12] and illustrated Fig. 2B and G (control in black line), distribution of AMPAR diffusion coefficients reveal two main populations centered approximatively at $0.8*10^{-2}\ \mu m^2\ s^{-1}$ (termed immobile trapped receptors) and $10^{-1}\ \mu m^2\ s^{-1}$ (termed mobile receptors). 30 min following NMDA treatment, we observed a 35% increase in the AMPAR mobile fraction (D coef > 0.02 $\mu m^2\ s^{-1}$) (t0: 30.11 ± 1.69%, t30: 40.65 ± 2.94%) (Fig. 2A–C). In contrast, we observed no change in AMPAR lateral diffusion with vehicle ($H_2O$) treatment (t0: 31.17 ± 1.94%, t30: 27.71 ± 2.98%) (Fig. 2D), or 30 min after an ATP treatment (t0: 31.19 ± 2.58%, t30: 30.44 ± 1.40%) (Fig. 2F–J). Similar results were obtained for synaptic trajectories of the GluA2-containing AMPAR (Supplementary Fig. 6).

This increase in AMPAR mobility following NMDAR-dependent LTD induction takes place progressively along the first 20 min following NMDA treatment (Fig. 2D) and remains stable up to 3 h (Fig. 2E; t0: 27.25 ± 3.41%, t180: 38.57 ± 2.56%). On the contrary, neither control nor ATP treatment induced such changes in AMPAR mobility (Fig. 2I, J). Moreover, in the presence of APV (50 μM), a specific NMDAR antagonist, no increase in AMPAR mobility was observed after NMDA treatment (Supplementary Fig. 7).

As previously described, AMPARs alternate between two main diffusion modes at the plasma membrane: an immobile one when trapped by interaction with scaffolding proteins, and a freely diffusive one[37]. We calculated the instantaneous AMPAR diffusion coefficients over time from individual synaptic trajectories[18]. For each trajectory, we classified AMPAR movement in three categories: receptors always mobile (class I), receptors always immobile (class II) and receptors alternating between mobile and immobile states (class III) (Fig. 2K). Two parameters were computed: the percentage of class II receptors, and the duration of immobilization of the class III receptors. The immobilization duration of mobile receptors, which reflects the avidity of AMPAR for trapping slots, was significantly decreased when LTD was induced by NMDA treatment (Fig. 2L). Before NMDA application, at synapses,

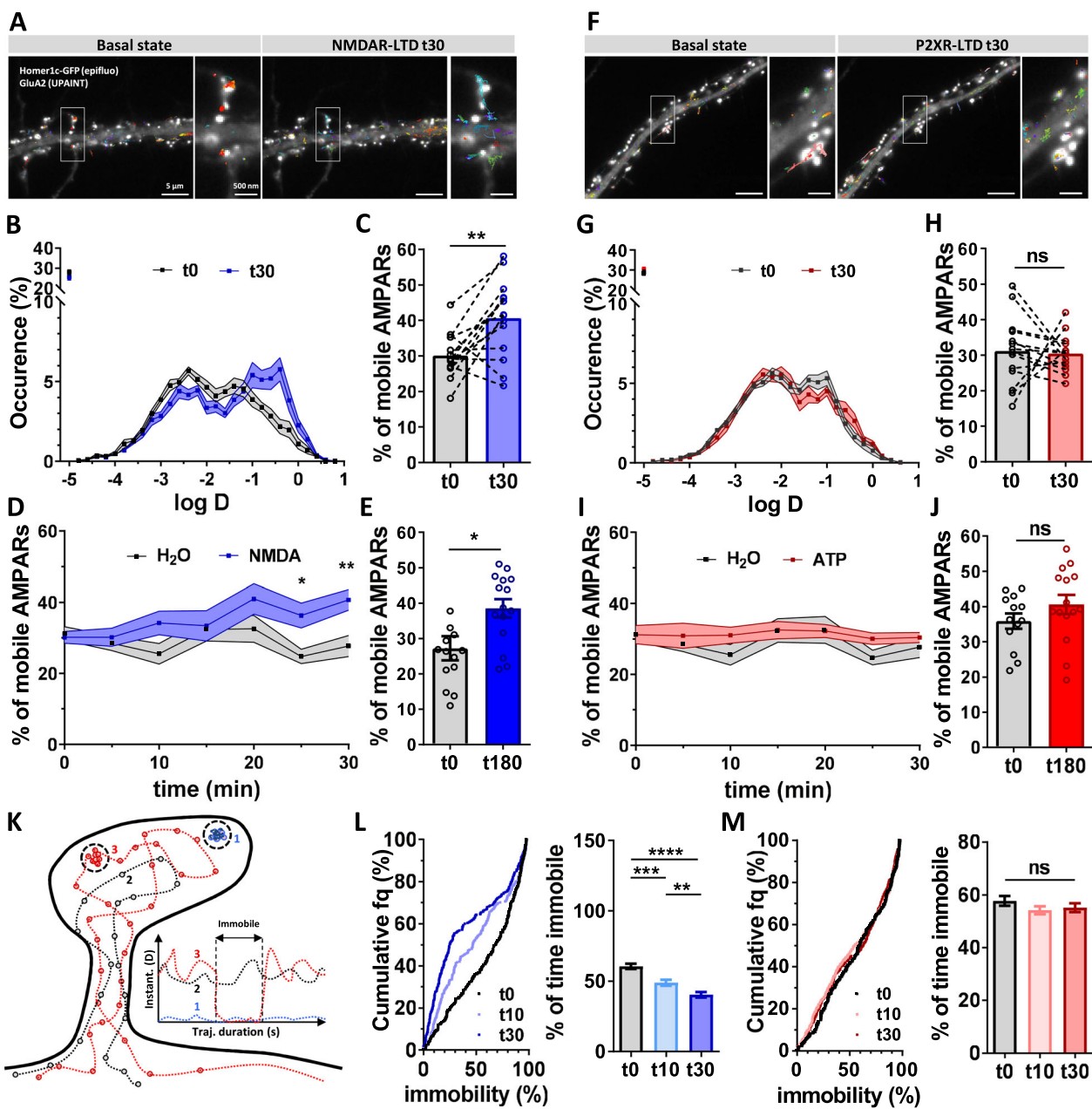

GluA2-containing receptors were immobile ~60% of the trajectory duration (60.56 ± 1.91) whereas this duration decreased to 48.92% ±2.11 after 10 min and to 40.37% ± 2.03 30 min following NMDA treatment (Fig. 2L). In contrast, after ATP treatment, this percentage remained unchanged (t0: 57.7% ± 1.83, t10: 54.14 % ± 1.54, t30: 55.13% ± 1.71) (Fig. 2M). Overall, these results demonstrate that NMDAR-dependent LTD, but not P2XR-dependent LTD, triggers an increase in AMPAR lateral mobility that takes place progressively after the LTD induction phase and remains stable for at least 3 h. At equilibrium, synapses depressed through NMDAR and P2XR thus harbor similar amounts of depression but distinct AMPAR mobility properties.

Given their supposed key role in LTD processes, we also characterized the endo/exocytosis properties at 3 h after LTD induction, when the long-term equilibrium of the depressed synapses is reached. We imaged neurons transfected with GluA1 and GluA2 labelled with pH sensitive supereclictic pHluorin (SEP) to measure directly their endocytosis (with the ppH protocol, Rosendale et al. 2017) and exocytosis rates[38–40], sampled for 5 min

for endocytosis or 1 min for exocytosis, 3 h after induction of LTD (Supplementary Figs. 4 and 5). We did not detect any modification of the frequency or amplitude of endocytic and exocytic events after either NMDA or ATP treatment compared to conditions before treatment or 3 h after vehicle application (H$_2$O) (Supplementary Figs. 4 and 5). This reveals that the maintenance of LTD is not due to a sustained modification of the endo/exocytosis balance of AMPARs.

In conclusion, the long-term equilibrium of depressed synapses seems to be maintained not by an endo/exocytosis unbalance but by a modification of AMPAR trapping at the PSD, increasing the pool of mobile receptors at the expense of the trapped ones.

**NMDAR-dependent LTD triggers a depletion of PSD-95 at synapses.** We next aimed at understanding the molecular mechanism underlying the changes in AMPAR mobility after the different LTD-inducing protocols. Various molecular modifications have been described to impact AMPAR mobility, including a diminution in the number of trapping slots, or a decrease in

**Fig. 2 NMDAR-dependent LTD but not P2XR-dependent LTD triggers a long-term increase of AMPAR lateral diffusion. A** Epifluorescence image of a dendritic segment expressing eGFP-Homer1c as a synaptic marker and GluA2-containing AMPAR trajectories acquired with uPAINT in basal state (left panel) and 30 min after NMDA treatment (right panel). **B** Average distribution of the log(D) (mean ± SEM), (D being the diffusion coefficient of endogenous AMPAR) in control condition (black line, $n = 14$) and 30 min after NMDA treatment (blue line, $n = 14$). **C** Average of the mobile fraction per cell, before and 30 min after NMDAR-dependent LTD induction ($n = 14$ cells, mean ± SEM, paired $t$-test, $p = 0.0042$). **D** Time-lapse (from 0 to 30 min) of GluA2-containing AMPAR mobility following NMDAR-dependent LTD induction (blue line) compared to vehicle application (green line) (mean ± SEM, $n = 14$ and 10 respectively). A significant increase of GluA2-containing AMPAR occurs 25 min after NMDA application. **E** Average histograms of the mobile fraction per cell, before and 180 min after NMDAR-dependent LTD induction ($n = 14$ and 15 cells, mean ± SEM, unpaired $t$-test, $p = 0.0123$). GluA2-containing AMPAR increased mobility remains stable for at least 3 h. **F–J** Similar experiments as from (**A–E**) has been realized with ATP-induced LTD protocol. **F** Epifluorescence image of a dendritic segment expressing eGFP-Homer1c with acquired trajectories of GluA2-containing AMPAR trajectories in basal state (left panel) and 30 min after ATP treatment (right panel). **G** Average distribution of the log (D) before (black line, $n = 14$) and 30 min (red line, $n = 14$) after ATP treatment (mean ± SEM). **H** Average of the mobile fraction per cell extracted from (**G**), ($n = 14$ cells, mean ± SEM, paired $t$-test, $p = 0.8234$). Contrary to NMDA-induced LTD, ATP-induced LTD is not associated with an increase of AMPAR mobility. **I** Time-lapse (from 0 to 30 min) of AMPAR mobility following P2XR-dependent LTD induction (red line) compared to vehicle application (green line) (mean ± SEM, $n = 14$ and 10 respectively). **J** Average histograms of the mobile fraction per cell, before and 180 min after P2XR-dependent LTD induction ($n = 13$ and 15 cells, mean ± SEM, unpaired $t$-test, $p = 0.1950$). No modification of AMPAR mobility is observed all along the 3 h experiments. **K** Scheme of the various AMPAR trajectory behaviors. AMPAR can be fully immobile (1, blue line), fully mobile (2 dark line) or alternate between mobile and immobile (3, red line). Calculation of the % of immobility all along the trajectory duration give an indication of the avidity of AMPAR for their molecular traps. **L** Variation of the % of AMPAR mobility per synaptic trajectories after NMDAR treatment (control (black line), 10 min (light blue line) and 30 min (dark blue line)). The left panel represents the cumulative distribution and the right panel the mean ± SEM. ($n = 252$, 235 and 280 synaptic trajectories respectively, one-way ANOVA $p < 0.0001$ and Tukey's post-test found significant differences $p = 0.0002$ and $p < 0.0001$ between t0 and t10, and t0 and t30 respectively, and $p = 0.0081$ between t10 and t30). **M** Variation of the % of AMPAR mobility per synaptic trajectories during ATP-induced LTD (control (black line), 10 min (light red line) and 30 min (dark red line) following LTD induction). The left panel represents the cumulative distribution and the right panel the mean ± SEM ($n = 264$, 434 and 326 synaptic trajectories respectively, one-way ANOVA $p = 0.3360$). Scale bars (**A** and **F**): 5 µm, and 500 nm for the zoom image on synapses.

AMPAR affinity for these slots by modifications in the AMPAR complex composition or phosphorylation status[32,41]. PSD-95 is the main scaffolding protein of the excitatory post-synaptic density and a major actor in AMPAR stabilization at synapses[7,42]. Hence, we used dSTORM to measure PSD-95 nanoscale organization following both NMDAR- and P2XR-dependent LTD induction (Fig. 3). As previously described, PSD-95 presents two levels of enrichment at synapses;[12,13,43] the first one delineating the PSD, the second one corresponding to small domains of local over-concentration into the PSD, beneath the AMPAR nanodomains and facing the glutamate release sites (Fig. 3A)[10–12]. Using tesselation-based clustering analysis[44], we extracted the first level (termed PSD-95 clusters), and the second level of clustering (termed PSD-95 nanoclusters) (Fig. 3A). After NMDA treatment, both PSD-95 clusters and nanoclusters displayed a decrease in number of PSD-95 (estimated number of PSD-95 per clusters (mean of the median per cell): t0: 114.7 ± 11.6, t10: 82.54 ± 10.99, t30: 66.93 ± 6.88; per nanoclusters t0: 28.12 ± 2.79, t10: 19.91 ± 2.02, t30: 18.99 ± 2.36) (Fig. 3B and C). We also observed a slight decrease in the nanocluster diameter (Supplementary Fig. 8). In contrast, following ATP treatment, the number of PSD-95 per clusters or nanoclusters remained unchanged (number of object per PSD-95 clusters t0: 112.5 ± 11.25, t10: 104.2 ± 9.85, t30: 102.4 ± 9.47; per PSD-95 nanoclusters t0: 23.7 ± 2.11, t10: 25.13 ± 3.25, t30: 21.42 ± 1.79) (Fig. 3D and E) as their overall organization (Supplementary Fig. 8).

Altogether, these results indicate that NMDAR-dependent LTD is associated with a change in PSD-95 number and cluster size, which is not the case for P2XR-dependent LTD. This could underlie the observed long-lasting increases in AMPAR mobility in the late phase of NMDAR-dependent LTD by decreasing the AMPAR trapping sites.

**Autophagosome-dependent degradation of the T19-phosphorylated form of PSD-95 is required for NMDAR-dependent LTD.** We investigated the molecular mechanism responsible of the PSD-95 loss when LTD is induced by NMDAR activation. The T19 residue of PSD-95, a GSK3β-phosphorylation

site, has been reported as crucial for the induction of LTD[45]. Thus, we investigated the impact of NMDAR-dependent LTD on both the AMPAR and PSD-95 reorganization, when expressing wild type or the phospho-null T19A mutant of PSD-95 (Fig. 4A and B). The overexpression of WT PSD-95 triggered an increase in both synaptic AMPAR and PSD-95 molecules per nanodomain correlated with an increase of mEPSC amplitude (mEPSC amplitude (pA), Ctrl: 12.33 ± 0.53, WT PSD-95: 15.75 ± 1.0) as previously reported[46,47]. The induction of NMDAR-dependent LTD on neurons expressing WT PSD-95, led to a decrease of the number of synaptic AMPARs and PSD-95 (Fig. 4A) mirrored by a decrease in mEPSC amplitude (Fig. 4B) (estimated number of AMPAR per nanodomain, WT PSD-95: 28.77 ± 0.83, WT + NMDA: 23.61 ± 1.13; estimated number of PSD-95 per cluster, WT PSD-95: 173 ± 8.82, WT + NMDA: 128.4 ± 9.37, mEPSC amplitude (pA), WT PSD-95: 15.75 ± 1.0, WT + NMDA: 12.15 ± 0.74). In contrast, the overexpression of PSD-95 T19A blocked the NMDA-induced decrease in synaptic AMPAR and PSD-95 amounts (Fig. 4A), and suppressed the depression of the mEPSC (Fig. 4B) (estimated number of AMPAR per nanodomain, T19A PSD-95: 25.01 ± 0.80, T19A + NMDA: 29.15 ± 1.05; estimated number of PSD-95 per cluster, T19A PSD-95: 145.5 ± 5.96, T19A + NMDA: 148.6 ± 8.07, mEPSC amplitude (pA), T19A PSD-95: 12.21 ± 0.52, T19A + NMDA: 13.86 ± 0.69) (Fig. 4A–C). As a control, none of WT and T19 A PSD-95 overexpression altered the LTD induced by ATP treatment (Supplementary Fig. 9A).

As the LTD-dependent synaptic removal of PSD-95 correlates with the increase in the diffusive fraction of AMPAR, we questioned the effect of T19A PSD-95 expression on the AMPAR lateral mobility (Fig. 4D). By using uPAINT on GluA2-containing receptors, we observed that neurons overexpressing WT PSD-95 present an increase in AMPAR mobility after NMDAR-dependent LTD induction (Mobile/immobile ratio, WT PSD-95: 0.24 ± 0.03, WT + NMDA: 0.40 ± 0.04), similar to the control (Fig. 2A–C). In contrast, the overexpression of PSD-95 T19A occluded the NMDA-induced increase of AMPAR mobility (Mobile/immobile ratio, T19A PSD-95: 0.23 ± 0.04, T19A + NMDA: 0.25 ± 0.03).

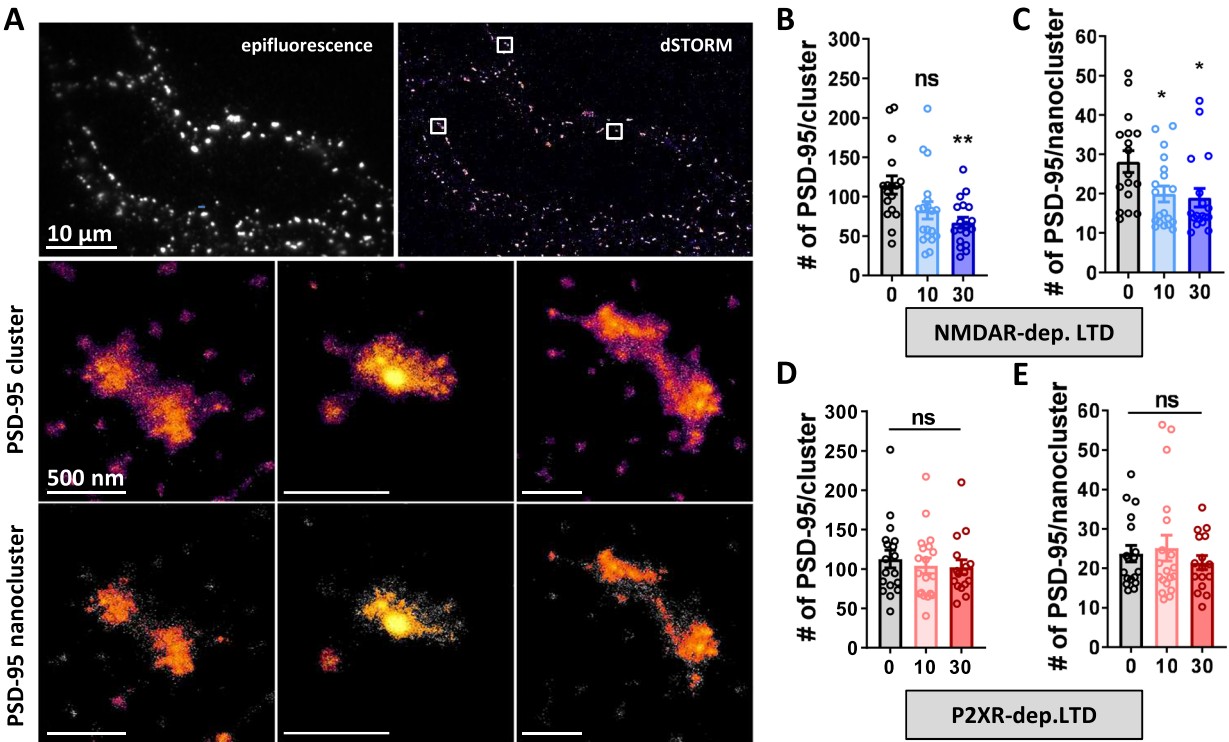

**Fig. 3 PSD-95 nanocluster organization is modified during NMDA- but not ATP-dependent LTD. A** Example of endogenous PSD-95 organization along a dendritic shaft observe with epifluorescence (top left) or obtained with dSTORM (top right), represented with SR-Tesseler software, Scale bars = 10 µm. Middle panels shows a PSD-95 clusters that have an enrichment factor above the average density factor (density color coded from magenta to yellow). Down panels shows PSD-95 nanoclusters within PSD-95 cluster in the middle panels, which corresponds to a PSD-95 structure with a higher density factor than the average PSD-95 cluster's density. Scale bars for middle and bottom images (PSD-95 clusters and nanoclusters) = 500 nm (**B** and **C**) Average number of PSD-95 molecules per cluster (**B**) and per nanoclusters (**C**) in basal state, 10 and 30 min after NMDA treatment (mean ± SEM, $n = 17$, 19 and 18 respectively, one-way ANOVA $p = 0.0059$ and Dunnett's post-test found significant between t0 and t30 conditions, $p = 0.0033$ but not between t0 and t10 conditions, $p = 0.0517$ for clusters; one-way ANOVA $p = 0.0189$ and Dunnett's post-test found significant between t0 and t10 and between t0 and t30 conditions, $p = 0.0348$ and $p = 0.0194$ respectively, for nanoclusters). **D** and **E** Average number of PSD-95 molecules per cluster (**B**) and per nanoclusters (**C**) in basal state, 10 and 30 min after ATP treatment (mean ± SEM, $n = 18$, 19 and 16 respectively, one-way ANOVA $p = 0.7616$ and $p = 0.5269$ for clusters and nanoclusters respectively).

We then tried to decipher the molecular mechanism which facilitates the removal of PSD-95 when phosphorylated at T19. Previous work has identified that PSD-95 co-immunoprecipitates with LC3, the protein responsible for autophagic cargo recruitment[48,49]. Moreover, recent work reported that autophagy is induced in dendrites by NMDAR-dependent LTD and is required for LTD induction[49]. Therefore, we directly tested if autophagy could be the degradation pathway of PSD-95 during LTD. To this end, we purified autophagic vesicles (AVs) from fresh hippocampal slices (Supplementary Fig. 10), before and 30 min after ATP- or NMDA-induced LTD, and determined the amount of both total and T19 phosphorylated PSD-95 in these vesicles (Fig. 5A and B). The induction of NMDAR-dependent LTD triggered a threefold increase of the T19-phosphorylated form and total amount of PSD-95 (total intensity of PSD-95, control: 0.71 ± 0.07, NMDA: 2.83 ± 0.35; and pT19-PSD-95, control: 0.62 ± 0.14, NMDA: 2.89 ± 0.77) in autophagosomes, whereas P2XR-dependent LTD did not affect their abundance in the purified vesicles (total intensity of PSD-95, control: 0.71 ± 0.07, ATP: 0.56 ± 0.1; and pT19-PSD-95, control: 0.62 ± 0.14, ATP: 0.57 ± 0.09) (Fig. 5C).

In parallel, we performed dual-color dSTORM experiments by labeling LC3, a typical marker of AVs, with alexa 647 nm, and PSD-95 with alexa 532 nm. A PSD-95 puncta was detected in the vast majority of AVs (43 out of 45 AVs, Fig. 5D). The quantification of PSD-95 signal reveals a threefold increase after

LTD induction compare to control (PSD-95 intensity (a.u), ctrl: 87.47 ± 24.33, NMDA: 301.6 ± 45.26).

In regard of these results, we hypothesized that NMDAR-dependent LTD triggers the activation of the GSK3β which phosphorylates PSD-95 at T19 to target it to autophagosomes for degradation. To validate this hypothesis, we first measured the evolution in function of time of AMPAR miniature currents following NMDAR-dependent LTD induction in the presence of TDZD8 (10 µM), an inhibitor of GSK3β (Fig. 5E). In the first 10 min, NMDA treatment triggered a classical decrease of synaptic strength, despite the presence of TDZD8. However, 20 and 30 min after induction, the synaptic depression was abolished (mEPSC amplitude (pA), Ctrl: 15.44 ± 0.95, NMDA: 10.41 ± 0.52, NMDA + TDZD8 10 min: 11.23 ± 0.70, NMDA + TDZD8 20 min: 17.07 ± 1.16, NMDA + TDZD8 30 min: 16.69 ± 1.40). Application of TDZD8 alone did not affect miniature amplitude neither at 10 min nor at 20 or 30 min (Supplementary Fig. 9B). Then, we measured mEPSC amplitude 30 min after NMDA and ATP-induced LTD in the presence of SBI (0.5 µM), a blocker of autophagosome formation (Fig. 5F). SBI alone did not impact mEPSC amplitude, while it entirely blocked NMDAR-dependent LTD expression 30 min after NMDAR activation. In contrast, P2XR-dependent LTD was properly expressed in the presence of SBI (mEPSC amplitude (normalized to Ctrl), Ctrl: 1.0 ± 0.04, SBI: 0.87 ± 0.06, NMDA: 0.73 ± 0.05, NMDA + SBI: 0.99 ± 0.04, ATP: 0.66 ± 0.02, ATP + SBI: 0.65 ± 0.04; Fig. 5F). A similar effect

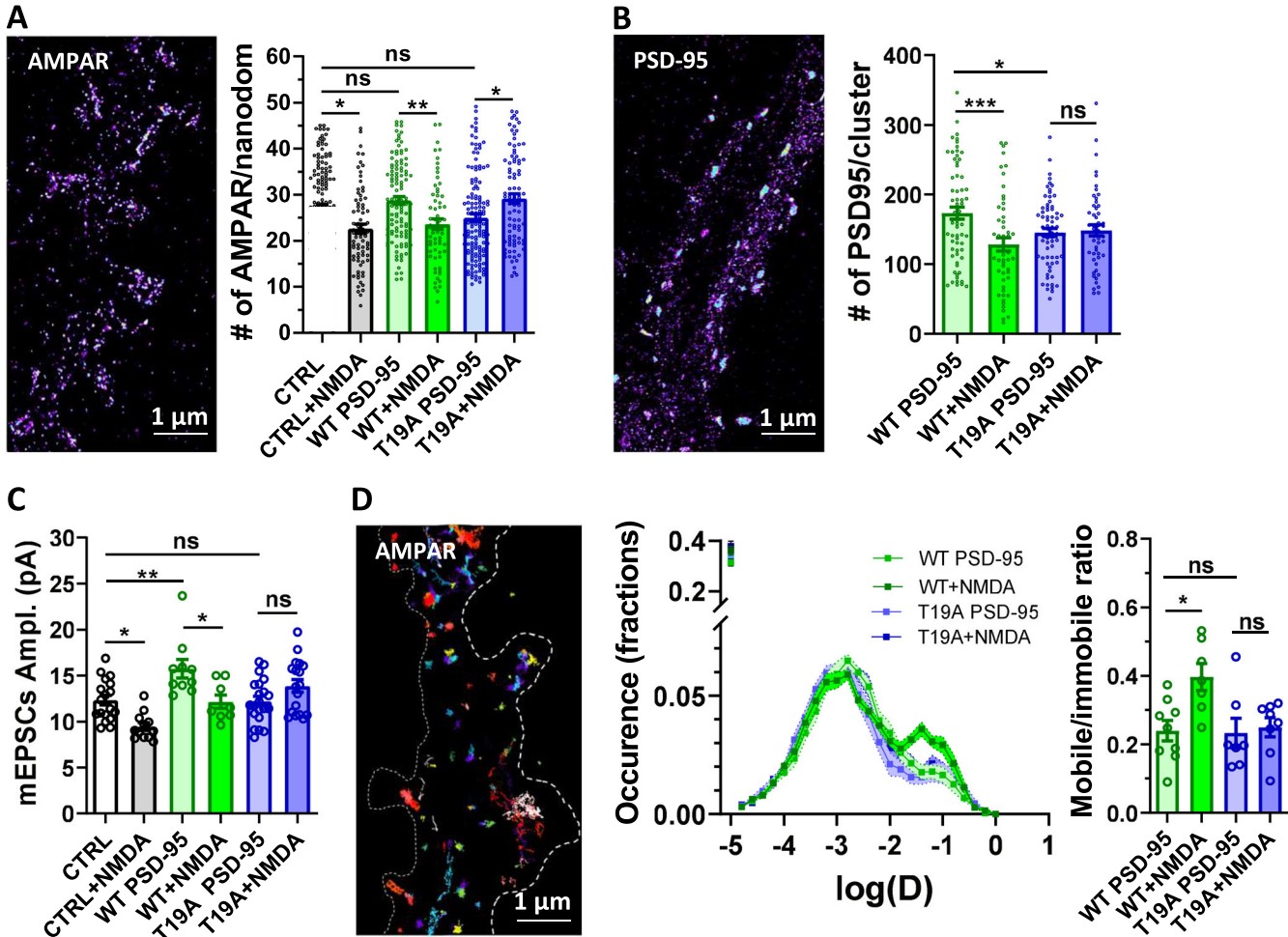

**Fig. 4 PSD-95 Phosphorylation at T19 position is essential for all the NMDAR-dependent molecular reshuffling induced by LTD. A** and **B** Expression of T19A phospho-null mutant of PSD-95, but not WT PSD-95, suppresses the decrease of GluA2 containing AMPAR (**A**) and PSD-95 (**B**) content per nanodomain 30 min following NMDAR-dependent LTD (at left: example of super-resolution intensity images of a piece of dendrite obtained using dSTORM technique). At right, the mean per cell histogram (mean ± SEM, one-way ANOVA, $p < 0.0001$ and $p = 0.0014$ respectively and Tukey's post-test results are realized between each conditions, $N = 141, 76, 110, 65, 139, 91$ for the measure of AMPAR per nanodomain, and $N = 65, 54, 70, 51$ for the measure of PSD-95 per cluster). **C** mEPSC amplitude is significantly decreased 30 min following NMDA treatment when both GFP or WT PSD-95 are expressed, while it is suppressed by T19A PSD-95 expression (mean ± SEM, one-way ANOVA, $p < 0.0001$ and Tukey's post-test results are realized between each conditions, $N = 18, 12, 10, 8, 21, 18$). **D** Example of trajectories of GluA2-containing receptors with uPAINT technique (left panel) and average distribution of the log(D) (middle panel) when WT (green lines) and T19A (blue lines) mutant PSD-95 are expressed, before (dark lines) and 30 min after (light lines) NMDA treatment. Average of the mobile fraction (Right panel), before and 30 min after NMDA treatment (mean ± SEM, one-way ANOVA, $p = 0.009$ and Tukey's post-test results are realized between each conditions, $N = 7, 9, 7, 8$). WT PSD-95 expressing neurons display an increase of AMPAR mobility following NMDAR-dependent LTD while T19A mutant expression abolished this mobility increase.

was obtained with the application of spautin-1 (10 μM), another inhibitor of autophagy (Supplementary Fig. 9C).

Overall, these experiments suggest that specifically NMDAR- but not P2XR-dependent LTD, triggers a phosphorylation at the position T19 of PSD-95, which targets PSD-95 proteins to autophagosomes where they are degraded. We propose that this suppression of PSD-95 is responsible of the maintenance of LTD and leads to an increase of AMPAR surface mobility.

**Short-term plasticity is increased during NMDAR-dependent LTD and requires AMPAR lateral diffusion**. We then aimed to determine if the difference in mobile AMPAR proportion at depressed synapses through NMDAR vs P2XR (Fig. 2) had any functional consequences on synapse function beyond their common decreased efficacy (Fig. 6). We have previously established that the pool of mobile AMPARs favors synaptic transmission during high frequency stimulation by allowing desensitized receptors to be

replaced by naïve ones[18,19,21]. In contrast, a decrease in the proportion of diffusive AMPAR, as triggered by artificial crosslinking[19], CaMKII activation[20] or by associating AMPARs with the auxiliary proteins TARPs[18], leads to a significant depression of synaptic transmission during rapid trains of stimulation. Therefore, we investigated whether the increase in AMPAR mobility observed during NMDAR-dependent LTD (Fig. 2B) could affect synaptic responses to high-frequency stimulus trains (5 pulses at 20 Hz) (Fig. 6A–C). We performed whole-cell patch-clamp recordings of CA1 neurons in acute hippocampal brain slices and measured short-term synaptic plasticity upon stimulation of Schaffer collaterals. LTD was induced by either ATP or NMDA treatment and paired-pulse responses were measured 30 min after LTD induction (Fig. 6A).

We first verified whether both ATP and NMDA treatment triggered a significant decrease in EPSCs amplitude. After NMDA application, we observed a ~35% decrease in evoked EPSC

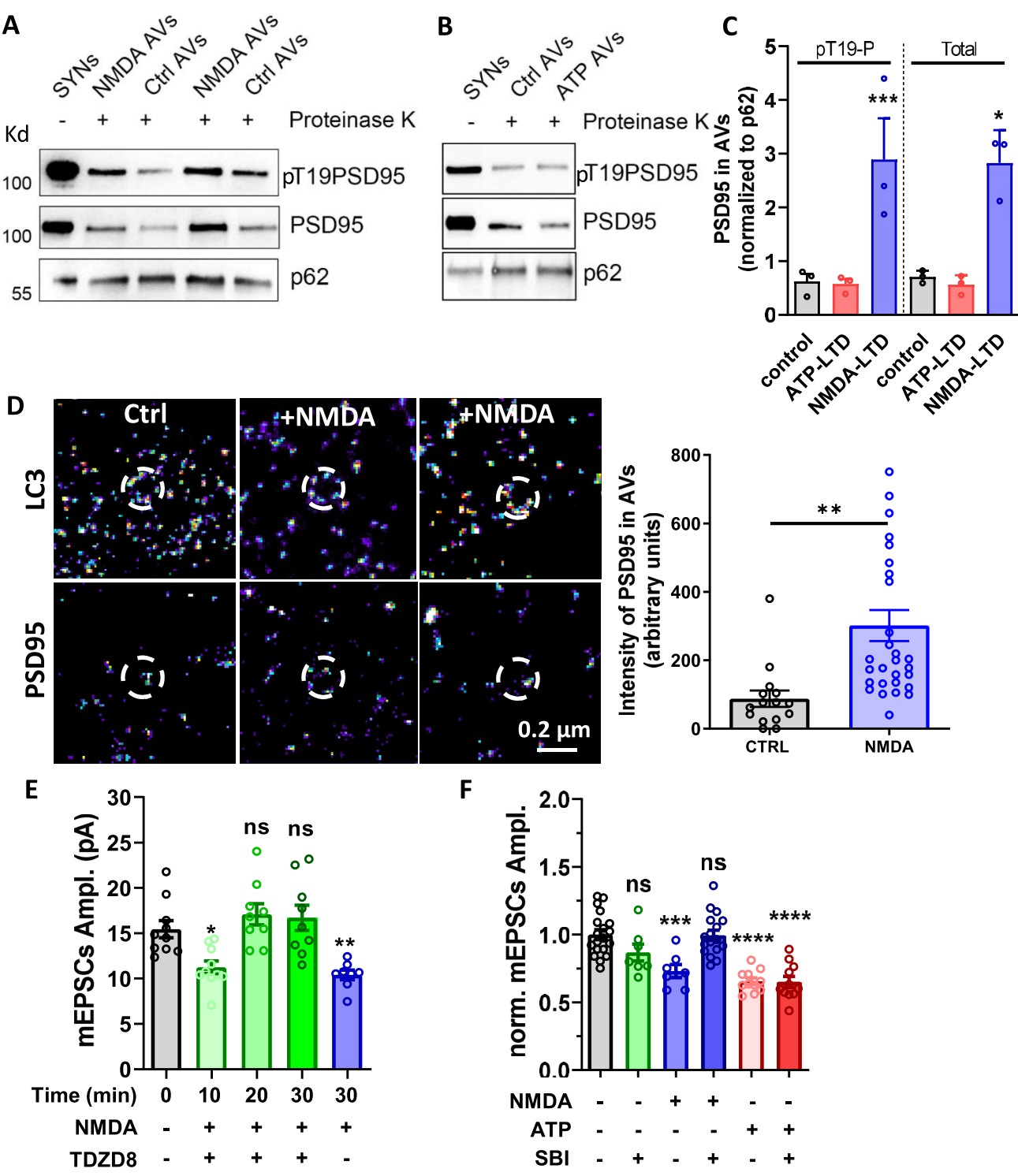

amplitude (basal state: $-37.63 \pm 2.48$ pA, 30′ NMDA: $-24.71 \pm 3.64$ pA) and ATP triggered a ~28% decrease in evoked EPSC amplitude (basal state: $-40.89 \pm 7.60$ pA, 30′ ATP: $-29.29 \pm 7.16$ pA) (Fig. 6A, B and D, E). We then analyzed paired-pulse responses. A representative trace and an average response are shown for NMDA (Fig. 6A) and for ATP treatment (Fig. 6D). Neurons expressing a P2XR-dependent LTD, which does not trigger an increase in AMPAR mobility, presented a paired-pulse

response similar to the one measured during the basal state (Fig. 6F). In contrast, neurons treated with NMDA, and thus exhibiting an increase in the proportion of mobile AMPARs, displayed a significant increase in the paired-pulse ratio compared to those at the basal state (Fig. 6C). This increase in the paired-pulse ratio was abolished when AMPAR were immobilized by antibody crosslinking (Supplementary Fig. 11), as previously described[18,19].

**Fig. 5 PSD-95 Phosphorylation at T19 position by GSK3β targets it to autophogosomes. A** and **B** Western blot analysis of total PSD-95 and T19 phosphorylated PSD-95 in purified synaptosomes and in PK-treated autophagic vesicles (AVs) purified before and after induction of the LTD by NMDA (A) or ATP (B) application. PSD-95 and T19PSD-95 levels were normalized to the levels of p62, an autophagic cargo. **C** Quantification of normalized T19PSD-95 and PSD-95 levels obtained in (**A** and **B**) reveals that both phosphorylated and global form of PSD-95 is over-accumulated in autophagic vesicles after NMDA treatment but not after ATP (mean ± SEM, n = 3, one-way ANOVA, p = 0.0005 for pT19P and p = 0.0180 for the Total. Dunnett's post-test results are realized between each condition and the control condition. For pT19P, control vs ATP-LTD p = 0.85 and control vs NMDA-LTD p = 0.0008; For Total, control vs ATP-LTD p = 0.996 and control vs NMDA-LTD p = 0.022). **D** Representative images of Dual-color dSTORM experiments with LC3 labelled with alexa-647 (upper panels) and PSD-95 labeled with alexa-532 (bottom panels). PSD-95 intensity inside the AVs shown a threefold increase following LTD induction by NMDA treatment (Right panel, mean ± SEM, unpaired t-test, p = 0.0022, n = 15 and 29). **E** Evolution in function of time of the mESPC amplitude after NMDAR-dependent LTD in the presence of TDZD8 (10 μM), an inhibitor of GSK3β. After 10 min, a normal LTD is induced but TDZD8 block the maintenance of the LTD (mean ± SEM, one-way ANOVA, p < 0.0001 and Dunnett's post-test results are realized between each conditions and the control condition, N = 10, 10, 9, 9, 8). **F** Average of the mESPC amplitude recorded on WT neurons 0 and 30 min after NMDA or ATP treatment, in the absence or the presence of SBI (a specific blocker of autophagy, 0.5 μm). SBI alone does not impact on mEPSC amplitude, while it fully blocks NMDAR-dependent LTD. At the opposite, ATP-induced LTD is preserved in the presence of SBI (mean ± SEM, one-way ANOVA, p < 0.0001 and Dunnett's post-test results are realized between each conditions and the control condition, N = 19, 7, 7, 17, 11, 11).

Short-term plasticity has traditionally been attributed to changes in pre-synaptic release probability[50], although it can also arise from AMPAR desensitization[15–17,19] and be regulated by AMPAR mobility[18–21]. To decipher between a pre- or post-synaptic origin of the NMDA-induced changes in short term plasticity, we directly measured the pre-synaptic probability of glutamate release before and after NMDAR- or P2XR-dependent LTD using the fluorescent glutamate reporter iGluSnFR[51]. We expressed iGluSnFR in cultured neurons and measured the variation in post-synaptic fluorescence upon triggering pre-synaptic action potentials by electrical field stimulations (Fig. 6G). None of the LTD protocols (ATP or NMDA) changed significantly the pre-synaptic release probability (Fig. 6H). These experiments indicate that NMDAR-dependent LTD favors the synaptic responsiveness to high-frequency stimulation through an increase in AMPAR mobility rather than a change in release probability.

Finally, we measured, in acute brain slices, the effect of synaptically induced NMDAR-dependent LTD, triggered by LFS, on paired-pulse response ratio. We first verified the proper LTD induction following LFS protocol as previously described[31]. After 1 Hz stimulation for 15 min, a 23 ± 5.1% decrease of the first peak amplitude was observed (EPSC amplitude, t0: −57.92 ± 3.96, LFS: −44.74 ± 4.03) (Fig. 6J). We then analyzed paired-pulse responses, a representative trace being shown before and after LFS (Fig. 6I). 30 min after LFS-induced LTD, neurons presented a significant increase in the paired-pulse ratio compared to before induction (PPR, t0: 1.43 ± 0.08 and LFS: 1.60 ± 0.09) (Fig. 6K), confirming that synaptically-induced LTD triggers a similar effect on synaptic responsiveness that NMDAR-dependent LTD.

**Modeling confirms that increasing AMPARs mobility improves synaptic responsiveness.** Both ATP- and NMDA-induced LTD resulted in a decrease in the overall AMPAR number at synapses. Moreover, single molecule tracking experiments (Figs. 2–4) show that LTD induced by NMDA, but not by ATP, is associated with an increase in AMPAR mobility. These changes in AMPAR diffusion can be related to a decrease in AMPAR complex affinity for their traps and/or a decrease in the number of synaptic traps (as reported by the decrease of total PSD-95 per synapses)[7,20,42,52,53].

To theoretically evaluate the impact of AMPAR endocytosis or untrapping on AMPAR mobility, organization and synaptic responses, we performed Monte-Carlo simulations using the MCell software (Fig. 7). The synaptic shape and perisynaptic environment were obtained from 3D electron microscopy images of hippocampal CA1 stratum radiatum area[10,54–56]. The simulation was divided in two sequences. The initial part simulates for 50 s, at the ms resolution, the dynamic

organization of proteins inside the synapse (Supplementary Fig. 12). The second part simulates for 250 ms, at the μs resolution, the AMPAR currents following 5 synaptic glutamate releases at 20 Hz (Fig. 7B to J, see methods). The protein properties, such as number and diffusion coefficient, were implemented into the model based on the results obtained with super-resolution imaging techniques and in agreement with previous papers[57], i.e., 200 PSD-95 and 120 AMPAR molecules (half in an internal pool, half at the surface, these values correspond to the mean of the mean per cell obtained in Supplementary Fig. 1 for AMPAR and Fig. 3 for PSD-95). Interactions between proteins were implemented following the scheme (Fig. 7A), and affinity constants (k) were adjusted to reach, at the equilibrium, a distribution similar to the one observed by microscopy (Supplementary Fig. 12).

Based on the literature and our experimental results, we tested the relation between simulated AMPAR current amplitudes and variations of two different interaction constants: (i) the endocytosis rate (k8*3), and (ii) the removal of PSD-95 into the PSD named PSD-95 inactivation rate (k6*4) (Fig. 7A).

A threefold increase in the endocytosis rate (noticed k8*3$_{Endo}$) which would correspond to the initial phase of both ATP- and NMDA-induced LTD, triggered a 25% decrease in the number of activated AMPAR (Fig. 7B). This value is similar to the current amplitude decrease measured with electrophysiology. In parallel, the increase in PSD-95 inactivation (k6*4$_{Inact}$) led to a 30% decrease in AMPAR current amplitude at the first glutamate release (Fig. 7C). Interestingly, this modification triggered a net increase in the number of mobile AMPAR (as illustrated in Supplementary Fig. 12B, blue line), mimicking the results observed during the late phase of NMDAR-dependent LTD (30 min and 3 h after LTD induction).

In another set of modeling experiments, we induced an AMPAR depletion into domains by increasing endocytosis rate (k8*3) and then put back this rate at its initial value. A rapid replenishment of nanodomains is observed (Supplementary Fig. 13A). Interestingly, this replenishment could be counter-balanced by an increase of AMPAR untrapping (k4*4) (Supplementary Fig. 13B).

We then determined the synaptic responses following trains of 5 stimulations at 20 Hz in these various conditions. k6*4$_{Inact}$ conditions triggered an increase in paired-pulse ratio, with a 18.7% increase in AMPAR activation for the second release and a 20.4% for the third one (Fig. 7C). While the k8*3$_{Endo}$ condition, which does not impact AMPAR mobility, did not modify the synaptic response in frequency (Fig. 7B).

These simulations demonstrate that using a realistic model of AMPAR organization, an increase in the pool of freely diffusing

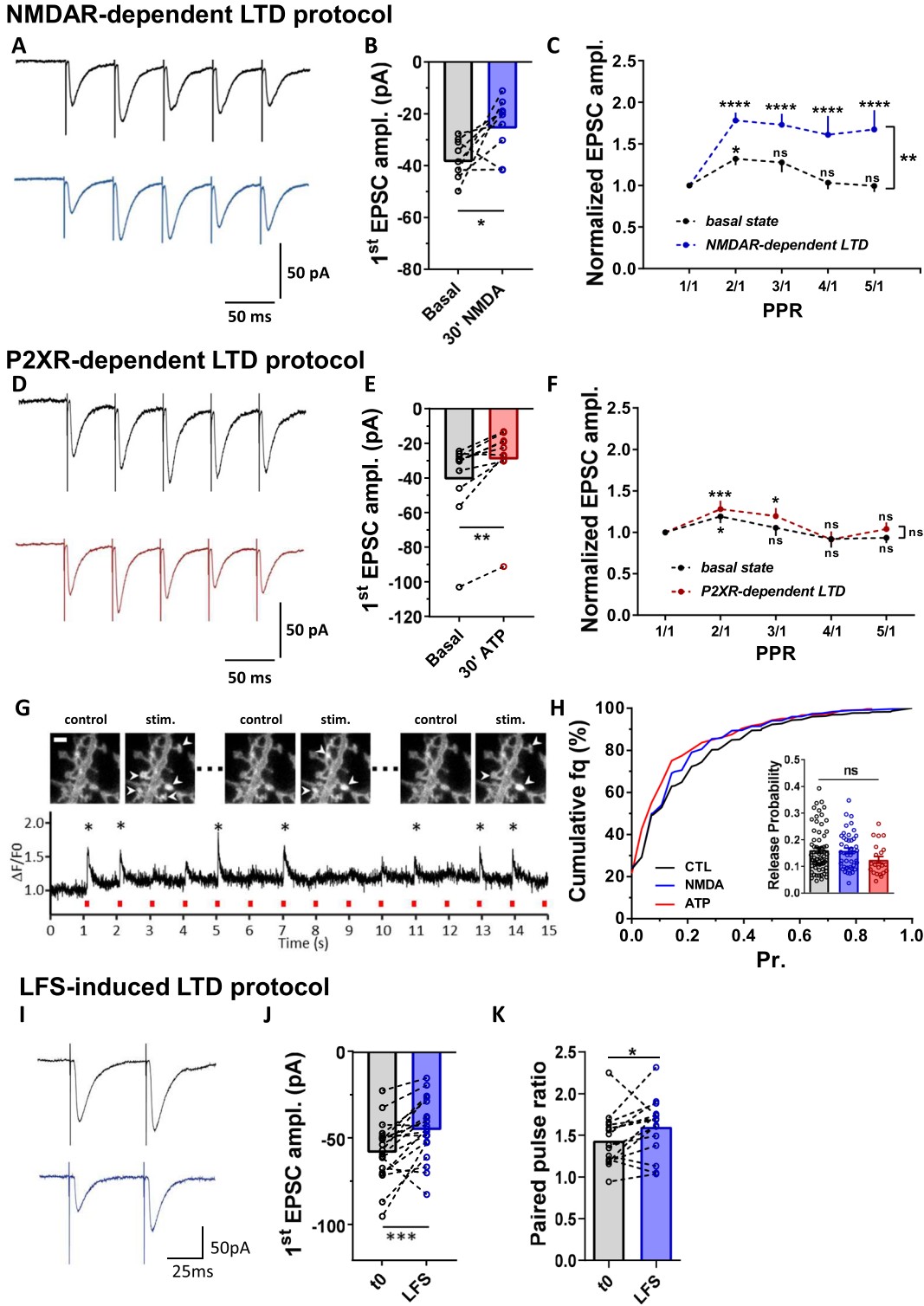

AMPAR induced by a decrease in the number of traps, is sufficient to trigger a paired-pulse facilitation similar to the one observed on brain slices by electrophysiological experiments.

Altogether, these experiments indicate that an increase in AMPAR mobility induced by the activation of NMDARs triggers an increase of synaptic responsiveness. Here we show that the plasticity paradigm (here the NMDAR-dependent LTD) regulates neuronal responsiveness through a post-synaptic mechanism that requires the degradation of PSD-95 by autophagosome and AMPAR surface mobility increases.

## Discussion

Using single molecule localization-based super-resolution microscopy on live and fixed neurons, combined with electrophysiology and modeling, we characterized the nanoscale modifications in AMPAR organization and dynamic triggered by two different types of LTD-inducing stimuli and estimated their impact on frequency-dependent synaptic current properties. We identified a common induction phase going through a depletion of AMPAR content both in nanodomains and at synapses, leading to a decrease in synaptic strength. A subsequent phase,

**Fig. 6 NMDAR-dependent LTD is associated to a frequency stimulation facilitation without affecting release probability. A** Representative traces of synaptic EPSCs in response to 5 stimulations at 20 Hz before (dark line) and 30 min after (blue line) NMDAR treatment. **B** Paired-average amplitude of the first response before and 30 min after treatment ($n = 9$ cells, mean ± SEM, paired $t$-test, $p = 0.0263$). The decrease of the first response demonstrates the efficiency of the LTD protocol. **C** Average of the 5 EPSC amplitudes, normalized by the first response intensity ($n = 9$ cells, mean ± SEM, two-way ANOVA. For PPR variation, $F(4,32) = 10.36$, $p < 0.0001$, Dunnett's post-test found significant differences increase of PPR between PPR1/1 and PPR2/1, $p = 0.0234$ at basal state, and between PPR1/1 and either PPR2/1, PPR3/1, PPR4/1 or PPR5/1, $p < 0.0001$, 30 min after NMDAR-dependent LTD induction. For basal state vs NMDAR-dependent LTD, $F(1,8) = 12.85$, $p = 0.0071$ and Sidak's post-test found significant difference between the basal state and 30 min after NMDAR-dependent LTD induction for PPR2/1, PPR3/1, PPR4/1 and PPR5/1, $p = 0.0011$, $p = 0.0013$, $p < 0.0001$ and $p < 0.0001$ respectively). A clear facilitation of the currents appears after induction of a NMDAR-dependent LTD. **D–F** Similar experiments has been realized when LTD is induced by ATP application, with example of traces in (**D**). The significant decrease of the first response represented in (**E**) validate the depression of the synaptic response ($n = 10$ cells, mean ± SEM, paired $t$-test, $p = 0.0012$). The average of the 5 responses (**F**) reveals no facilitation compared to control condition after ATP treatment ($n = 10$ cells, mean ± SEM, two-way ANOVA. For PPR variation, $F(4,36) = 7.73$, $p < 0.0001$, Dunnett's post-test found significant differences increase of PPR between PPR1/1 and PPR2/1, $p = 0.0163$ at basal state, and between PPR1/1 and PPR2/1 or PPR3/1, $p < 0.0004$ and $p = 0.0138$ for P2XR-dependent LTD. For basal state vs P2XR-dependent LTD, $F(1,9) = 1.197$, $p = 0.03023$). **G** Example of the fluorescence increase at a synapse expressing iGluSnFR construct during a field stimulation. Responding synapses are labelled with an arrow (upper part). At the bottom, example of the ΔF/F signal obtained at a single synapse. Stars indicate when the synapse is considered as stimulated (scale bar = 2 μm). **H** Cumulative distribution of the release probability per synapse in control condition (black line) or after LTD induction with either NMDA (Blue line) or ATP (red line) treatment. The mean values per recorded dendrites has been represented in the insert with the same color code. None of the conditions affects significantly the release probability ($n = 64$, 44 and 22 respectively, mean ± SEM, one-way ANOVA, $p = 0.1520$). **I** Representative traces of synaptic EPSCs in response to 2 stimulations at 20 Hz before (dark line) and 30 min after (blue line) LFS protocol. **J** Paired-average amplitude of the first response before and 30 min after treatment ($n = 19$ cells, paired $t$-test, $p = 0.0009$). The decrease of the first response demonstrates the efficiency of the LTD protocol. **K** Average of the paired-pulse ratio before and 30 min after LFS-induced LTD ($n = 15$; paired $t$-test, $p = 0.046$).

specific to NMDAR-dependent LTD, is associated with a net increase in the proportion of mobile AMPAR, and a depletion in PSD-95 clusters. The PSD-95 cluster modification is due to its phosphorylation at T19 position, driving it to autophagosome for degradation. Importantly, our experimental data and simulations using a realistic model indicate that this change in AMPAR dynamic allows synapses to improve their responsiveness to high frequency stimulations. Altogether, our data uncover an unexpected level of synaptic integration, where various LTD types do not similarly impact on synaptic molecular organization and function. This argues for a mechanism through which regulation of AMPAR surface density and diffusion following specific post-synaptic signaling to express LTD allows to adjust the capacity of synapses to encode pre-synaptic activity.

**P2XR- and NMDAR-dependent LTD are associated with nanodomain depletion proportional to the decrease in AMPAR mEPSC amplitudes.** Since the discovery that AMPARs are organized in nanodomains[12,13,43] and that AMPAR clusters are aligned with pre-synaptic release sites[10,11], our view of synapse function, and how it could be plastic, has strongly evolved. Together with previous findings[14,19,58,59], this has introduced the concept that post-synaptic plasticity could arise not only from absolute changes in AMPAR content, but also from their local reorganization regarding glutamate release site at the nanoscale. Thus, it became important to revisit LTD through this new prism. AMPAR-mediated current amplitude changes observed during LTD could arise from (i) a modification of the domain structure leading to a decrease in the packing of receptor;[14] (ii) a misalignment between the pre- and the post-synaptic machinery[10,59] or more simply (iii) a depletion of the domains and overall decrease in synaptic AMPAR content. Here we report that both NMDA and ATP treatments induce a synaptic depression based, at least partially, on post-synaptic nanoscale modifications. Super-resolution experiments revealed no modification in the overall nanodomain dimensions, and no pre-post synaptic misalignment. However, we observed a rapid (<10 min) depletion in the number of AMPAR per nanodomain. Interestingly, the extent of synaptic AMPAR depletion (around 30%) is proportional to the extent of mEPSC depression. This

suggests that synaptic depression can be mainly attributed to a decrease in post-synaptic AMPAR content per nanodomain.

**PSD-95 degradation through autophagy maintained the NMDAR-dependent LTD by decreasing the AMPAR trapping at domains.** The existence of various phases during NMDAR-dependent LTD has been described previously. A large increase in the AMPAR endocytosis rate after NMDA treatment has been reproducibly reported, but Rosendale et al. demonstrated that this increase is transient and goes back to its original value after ~10 min[34,35,60]. Sanderson et al. 2016, reported in parallel a rapid and transient increase in calcium-permeable AMPARs[61]. Modeling (Supplementary Fig. 13) reports that transient increase in endocytosis rate is able to induce but not to maintain a LTD. Because the consequence of endocytosis is to overfill the intracellular pool of receptors, the non-maintenance of this increased endocytosis rate should, if receptors are not degraded, lead to a replenishment of synapses by slow exocytose of these receptors (recycling). This means that the long-term maintenance of LTD requires additional mechanisms to the fast and transient increase in endocytosis. Endocytosis and exocytosis rates 3 h after LTD induction displayed a return to normal, demonstrating that neurons do not maintain a long-lasting unbalance between endocytosis and exocytosis to stabilize LTD. Modeling confirmed that the non-maintenance of this unbalance should lead to a replenishment of synaptic AMPAR and suppress the depression of synaptic currents, if receptors are not degraded. Here we observed a delayed increase in the proportion of mobile AMPAR, developing 20 min after NMDAR-dependent LTD induction. This mobility increase is correlated to a depletion of PSD-95 both inside the nanocluster and inside the entire PSD.

The expression of the T19A phospho-null mutant form of PSD-95 abolished both the PSD-95 reshuffling and the AMPAR increased mobility, favoring the hypothesis that the AMPAR mobility increase is directly due to the decrease in the number of their traps. Interestingly, inhibition of GSK3β during NMDAR-dependent LTD did not occlude the induction of LTD, as a ~27% decrease of miniature EPSC amplitude was observed 10 min after LTD induction in the presence of TDZD8. In contrast, inhibition of GSK3β blocked LTD maintenance (20 and 30 min after induction with NMDA).

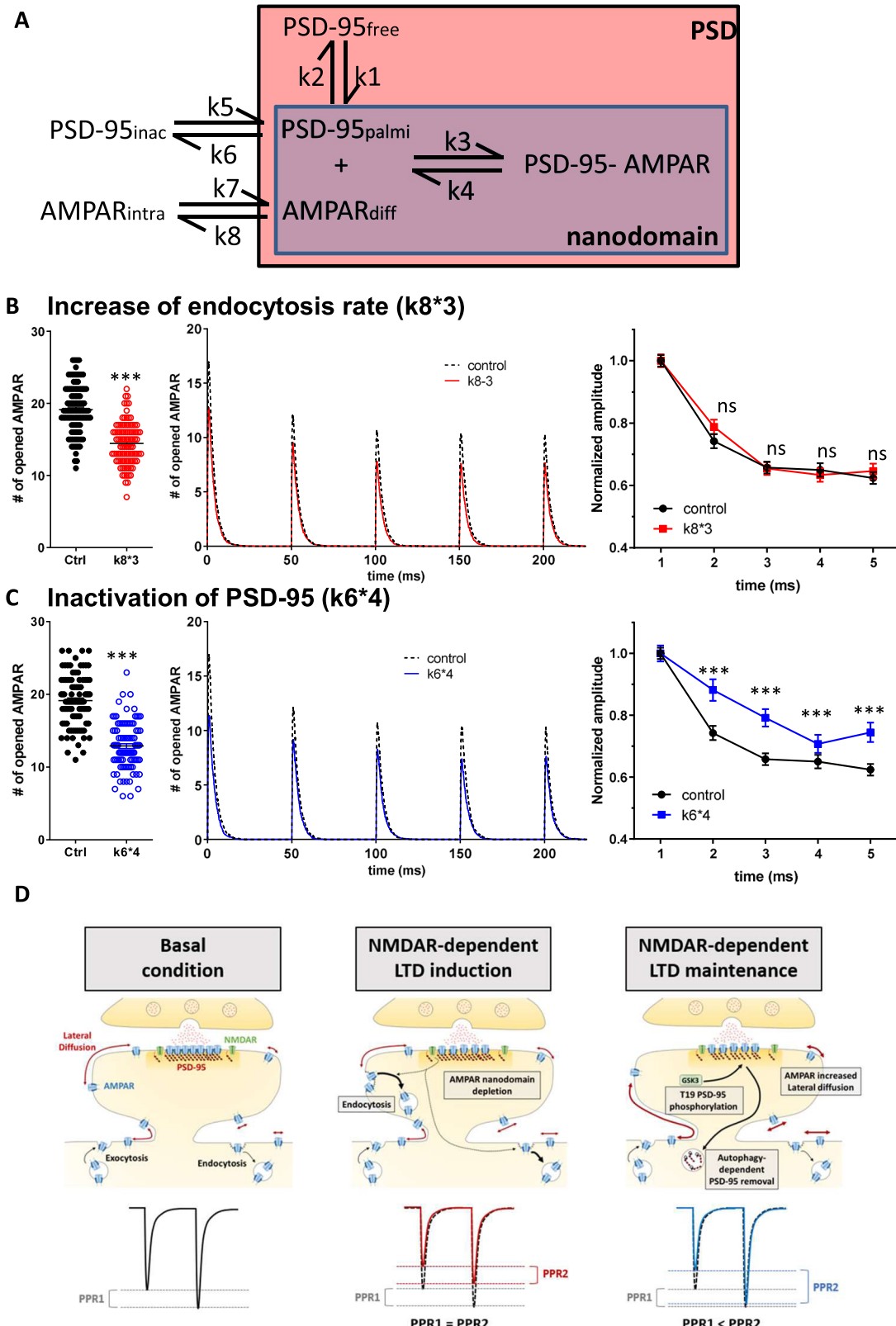

Previous work demonstrated how posttranslational modifications of PSD-95 can regulate its amount at the PSD and thus be important for LTD. As an example, PSD-95 ubiquitination by the E3 ligase Mdm2 removes it from the PSD and rapidly targets it to degradation by the proteasome[62]. Our results, combined to previous publications, allowed to sketch another quite complete scheme of the molecular mechanisms implicated in PSD-95 reshuffling during NMDAR-dependent LTD. Synaptic activity favors PSD-95 depalmitoylation[63] and activates GSK3β which phosphorylates PSD-95 at the T19 position[45]. Calcium bound calmodulin interacts with PSD-95 when phosphorylated at T19 and antagonizes palmitoylation, promoting release of PSD-95 out

**Fig. 7 In silico simulations confirm that AMPAR untrapping induced both a depression of synaptic currents and an increase of synaptic responsiveness. A** Representation of the main interactions which define the AMPAR organization/ trapping. PSD-95 can diffuse freely, be slowly mobile and confined into nanodomain when palmitoylated, or be inactivated. AMPAR can be endocytosed, freely mobile at the surface or being trapped by palmitoylated PSD-95 into the domain. Two various kinetic rate constant modifications trigger a synaptic depression, (1) an increase of endocytosis rate, mimicking the initial phase of the NMDAR-dependent LTD and the P2XR-dependent LTD (**B**). (2) A decrease of the total number of PSD through an increase of their inactivation rate (**C**), mimicking the depletion of PSD-95 observed during NMDAR-dependent LTD. For each condition, we report in the right panel, the number of open AMPARs during the first glutamate release (mean ± SEM), before (dark dots) and after (color dots) modification of the parameter. A significant decrease of AMPAR response similar to the depression experimentally measured is observed in all conditions. Middle panel, the average traces of the equivalent AMPAR current following 5 glutamate releases at 20 Hz. Right panel, the average of the AMPAR equivalent current following the 5 releases (mean ± SEM), normalized by the initial response. 96 independent simulations are realized in each condition. When depletion of synaptic AMPAR is induced by increasing the endocytosis, there is no modification of simulated paired-pulse ratio while PSD-95 inactivation condition triggers a significant increase of PPR. **D** Schematic summary of the molecular processes responsible of NMDAR-dependent LTD induction and maintenance. From basal state (left panel), NMDAR activation triggers an increase of endocytosis rate, responsible of the initiation of the depression (middle panel). Then the activation of the GSK3β phosphorylates PSD-95 at T19, targeting it to autophagosomes for degradation. This decrease of traps releases AMPAR out of the PSD, increase the amount of mobile receptors and favoring synaptic responsiveness (left panel).

of the PSD[64–66]. PSD-95 is then targeted to autophagosomes for degradation. This decrease in PSD-95 amount at the PSD limits the trapping of AMPAR, maintaining the depression after the initial induction phase. Regarding PSD-95, previous papers hypothesized that the role of autophagy during NMDAR-dependent LTD was to degrade AMPARs[48,49]. Here we demonstrate that during NMDAR-dependent LTD, there is also an autophagy of PSD-95 favoring the untrapping of AMPARs, which is not observed during P2XR-dependent LTD.

**The AMPAR mobility increase induced during NMDAR-dependent LTD mediates an improvement in synaptic responsiveness.** Short term synaptic plasticity (STP) depends on many factors, including pre-synaptic transmitter release[50], post-synaptic AMPAR desensitization[15–17,19], and AMPAR surface diffusion[18–21]. The latter mechanism potentiates synaptic responses to sequential stimuli by allowing desensitized receptors to be exchanged by naïve ones, hence improving the rate of recovery of synaptic depression due to AMPAR desensitization. Strikingly, we observed that following NMDAR activation, either by direct NMDA application or by synaptic activation through LFS stimulation, but not ATP, the ability of synapses to follow high frequency stimulation is improved. This differential effect between NMDA and ATP on STP mirrors their effect observed on AMPAR mobility. Therefore, it is attractive to suggest that NMDAR-dependent LTD increases STP through the increase in AMPAR mobility. This hypothesis is supported by the observation that: (i) NMDA does not modify pre-synaptic release probability, (ii) the time course of STP potentiation parallels that of the NMDA-induced increase in mobility, i.e., both processes only happen 20–30 min after NMDAR-dependent LTD, (iii) blocking the NMDA-induced increase in AMPAR mobility by AMPAR X-linking prevents the potentiation of STP, (iv) P2XR-dependent LTD, which does not affect AMPAR mobility, does not modify STP and (v) modeling confirmed that increasing the AMPAR mobile pool by decreasing the number of traps, favors the ability of synapses to follow high frequency stimulation. Of note, a similar increase in the paired-pulse ratio upon an increase in the proportion of mobile AMPAR has already been observed upon digestion of the extracellular matrix[21]. Here, our data suggest that a physiological increase in AMPAR mobility, after a protocol that induces LTD, triggers an improvement in synaptic responsiveness through potentiation of STP.

The comparison between P2XR- and NMDAR-dependent LTD reveals that LTD is a generic term, which comprises various physiological consequences. By definition, they all correspond to a net decrease of the amplitude of post-synaptic currents, and they are both induced by a transient increase in AMPAR endocytosis. However, while ATP-induced LTD just scales down the overall synaptic transmission properties, the NMDAR-dependent LTD affects more drastically the entire synaptic physiology. Indeed, the reshuffling of PSD-95 and AMPAR nanoscale organization induced by NMDAR-dependent LTD does not solely correspond to a decrease in the synaptic response amplitude but to deeper changes which modify the capacity of the depressed synapses to encode pre-synaptic inputs.

## Methods

**Hippocampal neuron culture and transfection.** The experimental designs and all procedures were in accordance with the European guide for the care and use of laboratory animals and the animal care guidelines issued by the animal experimental committee of Bordeaux Universities. Primary hippocampal cultures were prepared from E18 rat embryos of Sprague-Dawley rats according to the Banker protocol[67]. Briefly, hippocampi were dissected in Petri dishes filled with HBSS and HEPES, and dissociated by trypsin treatment (0.05%; Gibco) at 37 °C. For uPAINT experiments, neurons were electroporated (4D-Nucleofector system, Lonza, Switzerland) just after dissection with eGFP-Homer1c. 4 poly-L-lysine pre-coated 1.5H 18 mm coverslips were introduced in 60 mm dishes, which were pre-plated with 75,000 non-electroporated cells. Then, each dish was plated with electroporated neurons at the density of 250,000. After 2 h, coverslips were transferred to dishes containing an astrocyte feeder layer, plated at a density of 40,000 cells and cultured in MEM (Fisher scientific, cat No. 21090-022) containing 4.5 g/l Glucose, 2 mM L-glutamine and 10% horse serum (Invitrogen) for 14 days. Neuron cultures were maintained in Neurobasal medium supplemented with 2 mM L-glutamine and 1X NeuroCult SM1 Neuronal supplement (STEMCELL technologies) at 37 °C and 5% CO2, for 14–16 days.

Banker neurons were transfected with WT and T19A mutant of PSD-95, as well as GFP and SEP-GluA1 and SEP-GluA2 plasmids via calcium phosphate protocol (described in[10]).

**Sample preparation and immuno-labeling.** For dSTORM imaging of GluA2-containing AMPARs, primary neuronal cultures were treated with 30 μM NMDA (Tocris) for 3 min[31] or with 100 μM ATP (Sigma-aldrich) for 1 min in presence of CGS15943 (3 μM)[29,32]. After 10 or 30 min, neurons were incubated with a monoclonal mouse anti-GluA2 antibody (mouse antibody, diluted 1/100, provided by E. Gouaux, Portland, USA)[10,12,18] for 7 min at 37 °C and then fixed with 4% PFA. Then, cells were washed three times for 5 min in 1× PBS. PFA was quenched with NH4Cl 50 mM for 10 min. Unspecific staining was blocked by incubating coverslips in 1% BSA for 1 h at room temperature. Primary antibodies were revealed with Alexa 647 coupled anti-mouse IgG secondary antibodies (Thermo-Fisher, A21235).

For dSTORM and confocal imaging of PSD-95, primary neuronal cultures were treated either with 30 μM NMDA (Tocris) for 3 min or with 100 μM ATP (Sigma-aldrich) for 1 min and fixed with PFA 10 or 30 min after. PFA was quenched with NH4Cl 50 mM for 10 min. A permeabilization step with 0.2% triton X100 for 5 min was performed. Cells were washed three times for 5 min in 1× PBS. After three washes with 1× PBS, unspecific staining was blocked by incubating coverslips in 1% BSA for 1 h at room temperature. Cells were then incubated with monoclonal mouse anti-PSD-95 antibody (MA1-046, ThermoFischer), diluted in 1% BSA at 1/500, at room temperature for 1 h. Coverslips were rinsed three times in 1% BSA solution and incubated in 1% BSA for 1 h at room temperature. Primary antibodies were revealed with Alexa 647 (dSTORM) or Alexa 488 (confocal) coupled anti-mouse IgG secondary antibodies (ThermoFisher, A21235 and A11001).

Similar protocol is applied for LC3 labeling with incubation after permeabilization with polyclonal rabbit LC3 primary antibody (Sigma-aldrich, L8918, diluted at 1/500), revealed by anti-rabbit Alexa-647 nm antibodies.

For dSTORM imaging of pre- to post-synaptic alignment, primary neuronal cultures were incubated 0, 10 or 30 min after NMDA treatment with monoclonal mouse anti-GluA2 antibody[12] for 7 min at 37 °C and then fixed with 4% PFA. After permeabilization, unspecific staining was blocked by incubating coverslips in 1% BSA for 1 h at room temperature. Cells were then incubated with a polyclonal rabbit anti-RIM 1/2 antibody (synaptic systems, 140 203, diluted 1/200). Primary antibodies were revealed with Alexa 532 coupled anti-mouse IgG secondary antibodies (ThermoFisher, A21235) and with Alexa 647 coupled anti-rabbit IgG secondary antibodies (ThermoFisher, A21244).

**direct STochastic optical reconstruction microscopy (dSTORM)**. dSTORM experiments were done on fixed immunolabeled neurons. dSTORM imaging was performed on a LEICA DMi8 mounted on an anti-vibrational table (TMC, USA), using a Leica HCX PL APO 160 × 1.43 NA oil immersion TIRF objective and fibber-coupled laser launch (405 nm, 488 nm, 532 nm, 561 nm and 642 nm) (Roper Scientific, Evry, France). Fluorescent signal was collected with a sensitive EMCCD camera (Evolve, Photometrics, Tucson, USA). The 18 mm coverslips containing neurons were mounted on a Ludin chamber (Life Imaging Services, Switzerland) and 600 µL of imaging buffer was added[68]. Another 18 mm coverslip was added on top of the chamber to minimize oxygen exchanges during the acquisition to limit contact with the oxygen of the atmosphere. Image acquisition and control of microscope were driven by Metamorph software (Molecular devices, USA). Image stack contained typically 40,000–80,000 frames. Selected ROI (region of interest) had dimension of 512 × 512 pixels (one pixel = 100 nm).

The power of the 405 nm laser was adjusted to control the density of single molecules per frame, keeping the 642 nm laser intensity constant. Multicolor fluorescent microspheres (Tetraspeck, Invitrogen) were used as fiducial markers to register long-term acquisitions and correct for lateral drifts.

Super-resolution images with a pixel size of 25 nm were reconstructed using WaveTracer software[69] operating as a plugin of MetaMorph software.

**Cluster analysis**. AMPAR nanodomain analysis: localization of Alexa-647 signals was performed using PalmTracer, a software developed as a MetaMorph plugin by J.B. Sibarita group (Interdisciplinary Institute for Neuroscience). AMPAR nanodomain properties were extracted from super-resolution images corrected for lateral drift as described in previous studies[12,18].

PSD-95 cluster analyses: PSD-95 clusters and nanoclusters were then identified using SR-Tesseler software[44]. A first automatic threshold of normalized density DF = 1 was used to extract clusters of PSD-95 (cluster of level 1) having an enrichment factor higher than the average localization density, corresponding to Post-Synaptic Densities (PSD). A second threshold of DF = 1 applied on the localizations inside these clusters was used to identify the PSD-95 nanoclusters corresponding to domains.

AMPAR-RIM1/2 cluster distance measurement: localizations of Alexa-532 and Alexa-647 were corrected for chromatic aberration using a correction matrix calibrated from a set of tetraspeck beads imaged both with 642 nm and 532 nm excitation wavelengths. Clusters of AMPARs and clusters of RIM1/2 proteins were detected using the multicolor version of SR-Tesseler software[70] as described previously for single color SR-Tesseler, and distances between clusters detected in each color were measured within each synaptic ROI in order to solely measure the distances between objects belonging to the same synaptic contact.

**universal Point Accumulation Imaging in Nanoscale Topography (uPAINT)**. For u-PAINT experiments, the 18 mm coverslip containing neurons was mounted on a Ludin chamber (Life Imaging Services, Switzerland). Cells were maintained in a Tyrode solution equilibrated at 37 °C and composed of the following (in mM): 15 D-Glucose, 100 NaCl, 5 KCl, 2 MgCl2, 2 CaCl2, 10 HEPES (pH7.4; 247 mOsm). Imaging was performed on a Nikon Ti-Eclipse microscope equipped with an APO 100 × 1.49 NA oil immersion TIRF objective and laser diodes with following wavelength: 405 nm, 488 nm, 561 nm and 642 nm (Roper Scientific, Evry, France). A TIRF device (Ilas, Roper Scientific, Evry, France) was placed on the laser path to modify the angle of illumination. Fluorescence signal was detected with sensitive EMCCD camera (Evolve, Roper Scientific, Evry, France). Image acquisition and control of microscope were driven by Metamorph software (Molecular devices, USA). The microscope was caged and heated in order to maintain the biological sample at 37 °C.

The first step consisted to find a transfected neuron (eGFP-Homer1c, soluble GFP, soluble GFP + WT PSD-95 or soluble GFP + PSD-95 TA). This construct was used in order to visualize the neuron of interest and the synaptic area for more synaptic trajectory analysis. After selection of the dendritic segment of interest, ATTO647N coupled-anti-GluA2 antibody (mouse antibody, provided by E. Gouaux, Portland, USA) at low concentration was added in the Ludin chamber to sparsely and stochastically label endogenous GluA2-containing AMPARs at the cell surface. The TIRF angle was adjusted in oblique configuration to detect ATTO647N signal at the cell surface and to decrease background noise due to freely moving ATTO647N coupled antibodies. 647 nm laser was activated at a low

power to avoid photo-toxicity but allowing a pointing accuracy of around 50 nm, and 4000 frames at 50 Hz were acquired to record AMPAR lateral diffusion at basal state.

For LTD experiments lasting 30 min, chemical treatments to induce LTD were added into the Ludin chamber after the first movie acquisition. NMDAR-dependent LTD was induced using NMDA (Tocris Bioscience) at 30 µM for 3 min, while P2XR-dependent LTD was induced using ATP (Sigma-Aldrich) at 100 µM for 1 min in presence of CGS15943 (3 µM) as described in[29]. Imaging solution was washed and replace by fresh Tyrode solution and ATTO647N coupled-anti-GluA2 antibody at low concentration was added. A 4000 frames movie of the same dendritic segment was recorded at 50 Hz every 5 min for 30 min.

LTD experiments lasting 3 h were performed in non-paired conditions. Some coverslips were treated with the chemical compound inducing LTD while control are treated with water. Coverslips were placed into the culture dish at 37 °C and 5% CO2 and were imaged as previously described for uPAINT experiments. A single movie of 4000 frames at 50 Hz was acquired.

**Single-particle tracking analysis**. Single molecule localization, tracking and Mean Square Displacement (MSD) of ATTO-647N signals (uPAINT) were computed using PALMTracer software like in Nair et al 2013. From the MSD, two parameters were extracted: (i) the diffusion coefficient (D) corresponding to the global diffusion of the trajectory were calculated by linear fit of the first four points of the MSD plots. (ii) The instantaneous diffusion, corresponds to the variations of the D values all along the trajectory duration (see[18]).

**Biochemical purification of autophagic vesicles from hippocampal slices after NMDA and ATP treatment**. 200 µm-thick hippocampal sections were prepared from the brains of five C57BL/6 adult mice, using a vibratome (Leica, VT1200S), in the presence of ice-cold oxygenated a-CSF (124 mM NaCl, 74.55 mM KCl, 26 mM NaHCO3, 1.25 mM NaH2PO4, 10 mM glucose, 1 mM MgSO4 and 2 mM CaCl2). The sections were then incubated in a-CSF or were treated with an ATP pulse (100 µM for 1 min) or an NMDA pulse (50uM for 10 min), followed by a 30 min incubation in oxygenated a-CSF containing 10 nM of BafilomycinA1. After centrifugation for 2 min at 1.000 g at 4 °C, the supernatant was discarded and the pelleted sections were collected in 10 ml of 10% (w/v) sucrose, 10 mM Hepes and 1 mM EDTA pH 7.4, for homogenization using a glass homogenizer (20 dounces). Following a centrifugation for 2 min at 2.000 g at 4 °C, the post-nuclear supernatant was collected and AVs were purified as previously described[48,49]. Briefly, mitochondria and peroxisomes are removed with discontinuous Nycodenz gradients. The supernatant was placed on the top of the gradients and was centrifuged at 72,000 g for 1 h at 4 °C. The interface (Aps and endoplasmic reticulum) was isolated and diluted with an equal volume of HB buffer to be loaded on Nycodenz-Percoll gradients in order to remove the small-vesicular and non-membraneous material followed by a 51,000 g centrifugation for 30 min at 4 °C. The interface was collected and diluted with 0.7 V of 60% buffered Optiprep and the removal of Percoll silica particles followed by placing 8.5 ml of the diluted material in SW40 tubes overlayed with 1.5 ml of 30% iodixanol and a top layer of 2.5 ml of HB buffer. The material was then centrifuged at 51,000 g for 30 min at 4 °C resulting in the sedimented Percoll particles at the bottom of the tube and the autophagosomes band floated to the iodixanol/HB interface. Autophagosomes were collected for western blot analysis.

All fractions of the purification procedure were collected and analyzed by western blot with antibodies against the ER marker GRP78Bip (1:1000, ab21685), the nuclear marker Histone-H3 (1:1000, ab1791) and the autophagic vesicle LC3B-II (1:1000, L7543, Sigma). The purified vesicles were subjected to proteinase K (20 µg/ml) treatment for 20 min on ice to digest proteins associated with the outer membrane. Proteinase K was then inactivated with 4 mM PMSF for 10 min on ice and the material was centrifuged at 16.000 g for 20 min at 4 °C to pellet the AVs. The vesicles were lysed and boiled in laemmli buffer and analyzed by western blot using the following antibodies: a-PSD-95 (1:2000, MA1-046, Invitrogen), a-T19PSD-95 (1:1000, ab16496), a-p62 (1:2000, ab56416).

**Electrophysiological recordings**. mEPSC recordings in neuronal culture: coverslips of eGFP-Homer1c, soluble GFP, soluble GFP + WT PSD-95 or soluble GFP + PSD-95 T19A electroporated neurons were placed in a Ludin Chamber on an inverted motorized microscope (Nikon Eclipse Ti) and transfected neurons were identified under epifluorescence from the GFP signal. Extracellular recording solution was composed of the following (in mM): 110 NaCl, 5.4 KCl, 1.8 CaCl2, 0.8 MgCl2, 10 HEPES, 10 D-Glucose, 0.001 Tetrodotoxin and 0.05 Picrotoxin (pH 7.4; ~245 mOsm/L). For specific experiments, the extracellular solution was supplemented with an autophagy inhibitor, SBI (0,5 µM, Sigma-Aldrich) or GSK3β inhibitor, TDZD8 (10 µM, abcam). Patch pipettes were pulled using a horizontal puller (P-97, Sutter Instrument) from borosilicate capillaries (GB150F-8P, Science Products GmbH), and parameters are adjusted to reach a resistance of 4–6 MΩ. The pipettes are filled with intracellular solution composed of the following (in mM): 100 K-gluconate, 10 HEPES, 1.1 EGTA, 3 ATP, 0.3 GTP, 0.1 CaCl2, 5 MgCl2 (pH 7.2; 230 mOsm). Recordings were performed using an EPC10 patch-clamp amplifier operated with Patchmaster software (HEKA Elektronik). Whole-cell voltage clamp recordings were performed at room temperature and at a holding

potential of −70 mV. Unless specified otherwise, all chemicals were purchased from Sigma-Aldrich except for drugs, which were from Tocris Bioscience.

Miniature EPSC analysis were performed using a software developed by Andrew Penn, the matlab script is available on MATLAB File Exchange, ID: 61567; http://uk.mathworks.com/matlabcentral/fileexchange/61567-peaker-analysis-toolbox.

Paired-Pulse Response recordings in acute slices: acute slices were prepared from P16–18 Sprague-Dawley rats of both sexes. Rats were anesthetized with 5% isoflurane prior to decapitation according to the European Directive rules (2010/63/EU). Brain were quickly extracted and the two hemispheres were separated and placed in ice-cold, oxygenated (95% $O_2$,5% $CO_2$) sucrose-based artificial cerebrospinal fluid (ACSF) containing (in mM): 250 Sucrose, 2 KCl, 7 MgCl2, 0.5 CaCl2, 11 Glucose, 1.15 NaH2PO4 and 26 NaHCO3 (pH 7.4; ~305 mOsm/L). Sagittal slices were cut (350 μm thick) and incubated for 30 min at 32 °C in carbogenated ACSF (95% $O_2$,5% $CO_2$) containing (in mM): 126 NaCl, 3.5 KCl, 2 CaCl2, 1 MgCl2, 1.2 NaH2PO4, 25 NaHCO3 and 12.1 Glucose (pH 7.4; ~310 mOsm/L). Subsequently, slices were incubated for 30 min at room temperature and used until 5 h after preparation. Experiments were performed in a submerged recording chamber at 30–32 °C with continuous perfusion of carbogenated ACSF added with Gabazine (2 μM) and CGP52432 (2 μM). The intracellular solution was composed of (in mM): 130 Cs methane sulfonate, 10 HEPES, 10 EGTA, 2 MgCl2, 1 CaCl2, 4 Na2-ATP, 0.4 Na-GTP and 5 QX314. Synaptic responses were obtained by five stimulations of Schaffer collateral with 0.2 ms pulses at 50 Hz. 20 series spaced by 20 s were performed. LTD was induced by perfusion of NMDA (30 μM, 3 min), or ATP (100 μM, 1 min), in presence of CGS15943 3 μM. Another 20 series of 5 stimulations at 50HZ were performed 30 min after LTD induction or LFS (15 min 1 Hz stimulation at the Schaffer collateral). Average of each 20 series were calculated. Each response was normalized to the first one. Paired-Pulse Ratios were measured using Stimfit software taking into account fully successful paired-pulse response (trials with failures were rejected from analysis).

**iGluSnFR imaging and release probability measurement**. Transfection of iGluSnFR (Marvin et al., 2013) was performed on banker neuronal culture at 6 days in vitro (DIV) by a calcium phosphate transfection procedure. Experiments were carried out at 15–18 DIV. Thirty minutes after induction of LTD or water application for control, the neuronal preparation was placed under continuous perfusion in a Tyrode solution containing 100 mM NaCl, 5 mM KCl, 2 mM CaCl2, 2 mM MgCl2, 15 mM glucose, 10 mM HEPES pH 7.4. Experiments were performed at 35 °C on an inverted microscope (IX83, Olympus) equipped with an Apochromat N oil ×100 objective (NA 1.49). Samples were illuminated with a 473 nm laser (Cobolt) and emitted fluorescence was detected after passing a 525/50 nm filter (Chroma Technology Corp.). Images were acquired at a resolution of 100 × 100 pixels every 3 ms with a sCMOS camera (Prime 95B; Photometrics) controlled by MetaVue7.1 (Roper Scientific). Neurons were stimulated by electric field stimulation (platinum electrodes, 10 mm spacing, 1 ms pulses of 50 mA and alternating polarity at 1 or 10 Hz) applied by constant current stimulus isolator (SIU-102, Warner Instruments) in the presence of 10 μM 6-cyano-7-nitroquinoxaline-2,3-dione (CNQX) and 50 μM d,l-2-amino-5-phosphonovaleric acid (AP5) to prevent recurrent activity.

Image analysis was performed with custom-written macros in MATLAB (MathWorks) using an automated detection algorithm. At the end of the experiments, a 10 Hz stimulus was delivered for 5 s while images are acquired at 2 Hz in order to select only active synapses. A differential image was constructed by subtracting a five-frame average obtained immediately before the test train of stimulation from a five-frame average obtained just after stimulation. This difference image highlighting the stimulus-dependent increase of fluorescence was subjected to segmentation based on wavelet transform. All identified masks and calculated time courses were visually inspected for correspondence to individual functional pre-synaptic boutons. The mask was then transferred to the images acquired every 3 ms during a 1 Hz electrical stimulation. Successful fusion events were those where the fluorescence intensity of the first point following stimulation was greater than thrice the standard deviation of 200 points prior the increase in fluorescence. The measurement of release probability was made according to the number of successful responses over the total number of stimulations applied.

**Fluorescence imaging of exocytosis and endocytosis events**. Neurons (14–17 DIV, transfected at 7 DIV with SEP-GluA2 and SEP-GluA1) were perfused with HEPES buffered saline solution (HBS) at 37 °C. HBS contained, in mM: 120 NaCl, 2 KCl, 2 MgCl2, 2 CaCl2, 5 D-glucose and 10 HEPES, and was adjusted to pH 7.4 and 260–270 mOsm. For the ppH assay, MES buffered saline solution (MBS) was prepared similarly by replacing HEPES with MES and adjusting the pH to 5.5. All salts were from Sigma-Aldrich. HBS and MBS were perfused locally around the recorded cell using a 2-way borosilicate glass pipette. Chemical LTD was induced by incubating cells at 37 °C with either ATP (100 μM, 1 min) or NMDA (30 μM, 3 min) in culture medium, rinsed once and then incubated for further 3 h at 37 °C. When ATP treatment was used, cells were pre-incubated with CGS (3 μM, 10 min) to block adenosine receptors. For control experiments, water was used instead of ATP or NMDA.

Imaging was performed with an Olympus IX71 inverted microscope equipped for TIRF microscopy with a 150×, 1.45 NA objective (UAPON150XOTIRF), a

Laser source (Cobolt Laser 06-DPL 473 nm, 100 mW) and an Ilas2 illuminator (Gataca Systems) with a penetration depth set to 100 nm. Emitted fluorescence was filtered with a dichroic mirror (R405/488/561/635) and an emission filter (ET525/50 m, Chroma Technology) and recorded by an EMCCD camera (QuantEM 512 C, Princeton Instruments). Movies were acquired for 5 min at 0.5 Hz for endocytosis and for 1 min at 10 Hz for exocytosis. To achieve good signal/noise ratio required for event detection and further analysis, fluorescence was bleached by high laser power illumination prior to acquisition of the full movie. For imaging of exocytosis, this was done in the regular imaging buffer (HBS) to remove signal from fluorescent receptors localized at the plasma membrane or in non-acidic intracellular compartments[71] that could mask the appearance of newly exocytosed receptors. For imaging of endocytosis, bleaching was conducted while applying MBS at pH 5.5. At this pH, plasma membrane receptors are no longer fluorescent and protected from bleaching while intracellular receptors in non-acidic compartments are bleached (Rosendale et al. 2017).

Detection of both exocytic and endocytic events and their analyses was conducted using custom made Matlab scripts previously described[34,40], apart from kymographs which were obtained using ImageJ. Semi-automatic detection of endocytic events was performed as described previously[34]. In short, a sudden, punctate, fluorescence increase appearing in pH 5.5 images was detected as being an endocytic event if (1) it was visible for more than three frames (i.e., 8 s), and (2) it appeared at the same location as a pre-existing fluorescence cluster detectable in pH 7.4 images. Candidate events (768 events in 18 cells) were then validated by visual inspection in a random order to avoid any bias during cell stimulation (280 validated events, 48.87 + 5.94% per cell). This dataset was then used to train a support vector machine to validate the 47 remaining cells automatically to give 1447 validated events. Event frequency was expressed per cell surface area measured on the cell mask. Fluorescence quantification of events was performed as in[72]. In short, each value is calculated as the mean intensity in a 2-pixel radius circle centred on the detection to which the local background intensity is subtracted (the local background is taken as the 20–80th percentile of fluorescence in an annulus of 5–2 pixel outer and inner radii centred on the detection).

Semi-automatic detection of exocytic events was performed as described previously[40]. Fast fluorescence increases reporting exocytosis events were detected by generating a differential movie (imagen+1 - imagen + constant). A manual threshold was used to select candidate events (objects bigger than 2 pixels), with additional criteria to exclude moving clusters, variations in intense clusters, or tubule contraction. For each candidate event, a mini-movie and a series of background-subtracted images were generated. Events were validated or discarded by the user based on these two visualization tools. For fluorescence quantification, a ROI and a region surrounding the ROI (SR) are defined as follows. The five background-subtracted images before exocytosis define an SD of pixel values. A threshold is defined as seven times the SD to define putative ROIs. In case of multiple objects, the one closest to image center is chosen. If no object is detected, the ROI is defined as a 2.2 pixel radius circle centred on the centroid of the original detection. If an object is detected, the ROI is the reunion of the object and a 2.2 pixel radius circle centred on the object. The SR is obtained by a dilation of the ROI by two pixels. For the following frame, the same object detection procedure is applied, and a new ROI and SR are defined. The centroid of the new ROI must be <5 pixels away. If no object is defined, the ROI is kept the same. For images before exocytosis, the ROI used is the one defined at time 0, the time of exocytosis. For each event, we compute FR - S = FROI - FSR, where FROI and FSR represent the average fluorescence of the original images in the ROI and the SR, respectively. For the SR, the 20% lowest and highest pixel values are removed to limit environmental variations (out of the cell, bright cluster). For each event, FR - S is normalized by subtracting the average of values before exocytosis and divided by FR - S at the time of exocytosis. Events for which normalized FR - S was 50% for 2 s were sorted as burst events, and the other ones were sorted as display events.

**Modeling**. Computer modeling was performed using the MCell/CellBlender simulation environment (http://mcell.org) with MCell version 3.3, CellBlender version 1.1, and Blender version 2.77a (http://blender.org). The realistic model of glutamatergic synaptic environment was constructed from 3D-EM of hippocampal area CA1 neuropil as described in[54–56]. The 3D-EM reconstruction contains all plasma membrane bounded components including dendrites, axons, astrocytic glia and the extracellular space itself, in a 6 × 6 × 5 um3 volume of hippocampal area CA1 stratum radiatum from adult rat. The AMPAR chemical kinetic properties were obtained from the well-established model published in Jonas et al., 1993, and the kinetic parameters were adjusted to fit with the recorded mEPSCs (see[10]).

Three surface properties are defined. The synapse, the PSD (identified on EM data) and a 100 nm domain inside the PSD which correspond to the AMPAR nanodomain. According to literature and to our dSTORM data, 200 PSD-95 molecules were released. They freely diffuse inside the PSD and are palmitoylated at a certain rate (kon = 35, koff = 0.7) when they enter inside the nanodomain area. These kinetic rate constants result in a steady-state accumulation of around 70 palmitoylated PSD-95 inside the nanodomain according to experimental results. PSD-95 can be also inactivated with a certain rate to mimic LTD.

Concerning AMPAR, a total of 120 receptors were released at time zero and were distributed in two separate pools as follow: one pool of 60 AMPARs were allowed to diffuse on the membrane surface and a second pool of 60 AMPARs

represented the endocytosed state. Individual receptors exchange between the pools at equal forward and backward rates to maintain equilibrium. To mimic LTD induction, the rate of exchange to the endocytosed state was increased. At the surface, AMPARs are mobile with a diffusion coefficient of $0.5\,\mu m^2\,s^{-1}$, and can interact inside the nanodomain with palmitoylated PSD-95 (kon = 5, koff = 1). This interaction slows down the AMPAR to $0.005\,\mu m^2\,s^{-1}$ and slows down the AMPAR diffusion constant to retain the molecular complex within the nanodomain leading to an equilibrium of around 20–25 AMPAR trapped into the domain (similar to results described in the literature). All this organization of PSD-95 and AMPAR at the synapse were simulated at a time step of 1 ms for 50,000 iterations (50 ms), until reaching an equilibrium (as illustrated Supplementary Fig. 13). Then the simulations were switched to a time step of 1 µs for 250,000 iterations to model the AMPAR responses when the glutamate is released at the pre-synaptic level, in front of the nanodomain.

We tested three individual modifications of the kinetic rate constants to mimic LTD. (1) A multiplication by 3 of the endocytosis rate mimics the endocytosis-dependent LTD (ATP type or initial phase of NMDA type of LTD). (2) A four times increase of the PSD-95 + AMPAR koff, which favors the untrapping of AMPAR. (3) A four-time increase of the inactivation rate of PSD-95, mimicking the decrease of PSD-95 experimentally observed. 100 simulation trials are averaged for each condition, and 2500 glutamate are releases for each release event.

**Sampling and statistics**. All data showing AMPARs and PSD-95 organization, AMPAR lateral diffusion, PSD-95 autophagy and synaptic currents, AMPAR endocytosis are pooled from at least three independent experiments. Summary statistics are presented as mean ± SEM (Standard Error of the Mean). Statistical significance tests were performed using GraphPad Prism software (San Diego, CA). Normality tests were performed with D'Agostino and Pearson omnibus tests. For non-normally distributed data, we applied Mann–Whitney test or Wilcoxon matched-pairs signed rank tests for paired observations. When the data followed normal distribution, we used paired or unpaired t-test for paired observations unless stated otherwise. All statistics are two-sided tests comparing groups of biological replicates. ANOVA test was used to compare means of several groups of normally distributed variables. Indications of significance correspond to $p$ values < 0.05(*), $p$ < 0.005(**), and $p$ < 0.0005(***). After ANOVA analysis, we apply a Dunnett's post test to determine the p value between two conditions, results of these tests are noted Anova post-test.

**Ethical approval**. All experiments were approved by the Regional Ethical Committee on Animal Experiments of Bordeaux.

**Reporting summary**. Further information on research design is available in the Nature Research Reporting Summary linked to this article.

## Data availability
Source Data that support the findings of this study are available in the online version of the paper and from the corresponding author upon reasonable request.

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

## Acknowledgements

We acknowledge E. Gouaux for the anti-GluA2 antibody and J.S. Marvin for the iGluSnFR plasmid; the Bordeaux Imaging Center, part of the FranceBioImaging national infrastructure (ANR-10INBS-04-0, for support in microscopy; E. Normand and the IINS in vivo facility for animal husbandry. We thank the IINS cell biology core facilities (LABEX BRAIN [ANR-10-LABX-43]) and in particular C. Breillat, E. Verdier, and N. Retailleau for cell culture and plasmid production, and Jorge Aldana from the Salk institute for computing support. This work was supported by funding from the Ministère de l'Enseignement Supérieur et de la Recherche (ANR NanoDom and AMPAR-T), Fulbright and Philippe foundation to E.H. and D.C., Centre National de la Recherche Scientifique (CNRS), ERC grant ADOS (339541) and DynSynMem (787340) to D.C., Fondation pour la Recherche Médicale fellowship to B.C.

## Author contributions

B.C. and C.C. performed all dSTORM and single-particle tracking experiments, culture and slice electrophysiology. M.M. performed iGluSnfr experiments, D.P. and S.S. performed the endo/exocytosis experiments. E.H., T.M.B and T.J.S. created the MCell model and E.H. performed the simulations. C.B., A.K. and J.B.S. developed the super-resolution analysis software. R.V.K. and A.B.S. participate to conception and validation of some hypothesis. E.K. and V.N. realized biochemical experiments on autophagosomes. N.R. performed molecular biology constructs. E.H. conceived and supervised the study. D.C. financed the study. E.H., B.C. and D.C. wrote the paper. All authors contributed to the revision of the paper.

## Competing interests

The authors declare no competing interests.
