## [Peer Review File · Nature Communications]

Reviewer #1 (Remarks to the Author):

This study has the potential to advance our understanding of cellular and molecular mechanisms underlying LTD. In detail, LTD was chemically induced in cultured hippocampal neurons with either NMDA or ATP (in the presence of a blocker of adenosine receptors) to stimulate the NMDAR or P2X receptors, respectively. Both treatments reduced the number of AMPAR in postsynaptic nanodomains and, more generally, dendritic spines, and the amplitude of mostly AMPAR-mediated mEPSCs. NMDA but not ATP increases lateral mobility of GluA2-containing AMPARs. In parallel, PSD-95, the main anchoring protein for AMPAR at postsynaptic sites, also showed a reduction in localization to PSDs and its nanodomains therein following NMDA but not ATP treatment. These results are consistent with earlier findings and expand those by analyzing AMPAR content in nanodomains and AMPAR mobility.

Earlier work showed that LTD requires that PSD-95 is phosphorylated on Thr19 near its N-terminus by GSK3. The authors find that overexpression of PSD-95 with the T19A mutation but not of WT PSD-95 prevented the decrease in AMPARs in nanoclusters, of PSD-95 in PSDs and of mEPSC amplitude induced by NMDA and the increase in AMPAR diffusion. This is again consistent with earlier work.

Perhaps most interesting are data that suggest that WT but not T19A mutant PSD-95 would increase in autophagosomes following NMDA treatment. Application of a GSK3 inhibitor and of a blocker of autophagy prevented the NMDA-induced downregulation of mEPSC amplitude at the 30 min post NMDA treatment time point (although the GSK3 inhibitor did not prevent a temporary reduction in mEPSC at the 10 min time point).

Major Concern

My main concern is that the most interesting finding that PSD-95 needs to be targeted to autophagosomes for lasting LTD is not sufficiently supported.

1. Subcellular fractionation suggests that PSD-95 is targeted to autophagosomes. The purification of autophagosomes has to be described in detail, not just referenced to earlier work to ensure that there are no differences in the protocols. More importantly, the different fractions have to be more extensively characterized with various subcellular markers than currently provided.

2. At the same time colocalization of PSD-95 with autophagosomes has to be demonstrated by immunostaining, ideally with superresolution.

3. For testing whether the LTD they monitor by mEPSC analysis depends on autophagy, the authors use only a single inhibitor (SBI) of autophagy, which could act via side effects rather than by inhibiting autophagy. In fact, there is a tendency for SBI to decrease mEPSC amplitude under basal conditions by itself although it did not reach statistical significance. At least one more inhibitor that acts in a way that is different from SBI is required to be certain enough that autophagy is required for LTD.

4. The effect of TDZ8 alone has to be documented in Fig. 5E

Modest Concerns

5. It is slightly puzzling that mEPSC frequency was not reduced upon ATP treatment because the observed reduction in mEPSC amplitude should have shifted smaller mEPSC events below the detection threshold given by the electric noise. Perhaps the ATP paradigm effected mostly larger and not smaller mEPSC events. A hint of an analogous potential difference in effects on AMPAR content of larger versus smaller AMPAR nanodomains is perhaps seen in Fig. 1B versus 1H. A more detailed analysis of mEPSC frequency distribution might shed light onto this issue.

Minor Concerns

1. p. 8: the statement “PSD-95 can be immunoprecipitated by LC3” should be better worded, perhaps “PSD-95 co-immunoprecipitates with LC3.”

2. p. 13: “Calcium bound calmodulin interacts with PSD-95 when phosphorylated at T19...”
References here are not fully accurate and also not complete (the phosphorylation dependent Calcium / calmodulin binding was shown in Chowdhury et al 2017: EMBO J 37, 122-138, not in the cited paper).

3. Fig 5: the label “T19PSD95” would be clearer as “pT19 PSD95.”

Reviewer #2 (Remarks to the Author):

In a primary cell culture model of NMDA or ATP induced depression of synaptic transmission, the authors show that the two forms use different mechanisms to reduce synaptic AMPA receptors. While NMDA reduces PSD-95 and increases the mobility of AMPA receptors, ATP does not affect PSD-95 and AMPA receptor mobility. The author further show that overexpressing PSD-95 T19A impairs the NMDA mediated synaptic depression selectively and it is also impaired by pharmacological blockade of GSK-3 and autophagosome generation. This study is a continuation on a number of studies that aimed to establish that NMDA-dependent long-term synaptic depression is caused by the synaptic removal of PSD-95 as a slot for AMPA receptors. The novel finding here is that after NMDA treatment, AMPA receptor mobility increases and that this prevails throughout the expression of the depressed synaptic state. The authors corroborate this finding by imaging the diffusion of endogenous AMPA receptors with uPaint and alterations in short-term synaptic plasticity. While this finding is novel and interesting, its significance would rise from an agonist induced cell culture model to a potential mechanism of learning and memory, if the authors could show that it also applies to synaptically triggered NMDA receptor-dependent LTP.

Include different proposed mechanisms of PSD-95 removal from PSD for LTD as previously proposed in discussion and how they relate to the new proposed mechanism, including depalmitoylation, ubiquitination, calmodulin N-terminal masking and T19-dependent mobilization.

Text lacks reference to corresponding figures in multiple cases.

Fig.1 Scaling of mEPSC sample traces after NMDA or ATP appears smaller. Is the reduction calculated with correctly scaled responses? Revise. Also add representative traces in the following figures as based on the problem in Fig. 1, the data in the others cannot be judged.

Fig.5S1 Wrong title and incomplete legend.

109 here no pathway-specific modifications are shown

132 If AMPA receptor is your marker for the nanodomain and you have a decrease in AMPA receptor

numbers, how does the diameter not change? Does this mean that AMPA receptors are only removed from the center?

247 Specify that this mechanism applies only to NMDA. For ATP, the mechanism is not further explored.

291 Stein et al 2003 reported this earlier. Include earlier reference.

324 If the ratio of T19/total PSD-95 does not change in the AV, why would that be a specific signal to target them there? You would need to compare the T19/total ratio in AV versus synaptosomes for this conclusion.

417 these numbers do not match the experimental data of this study

Reviewer #3 (Remarks to the Author):

In this manuscript Compans et al. examine the mechanisms underlying NMDAR vs P2XR dependent LTD using cell biology, biochemistry and electrophysiology. In summary, they show that NMDA receptor and P2XR LTD both involve AMPAR endocytosis and reorganization of AMPARs in nanodomains in the postsynaptic membrane but they provide evidence that NMDAR but not P2XR regulates PSD-95 in the PSD by removal of T-19 phosphorylated PSD-95 by autophagosomes.

This manuscript contains a lot of data that provides evidence for these conclusions. First, using super resolution (dSTORM) imaging they show that both NMDAR and P2XR induced LTD results in the number of AMPAR in specific nanodomains with little change in the nanodomain size. This was associated with decreases in mEPSC amplitude as well. In contrast NMDAR but not P2XR induced LTD increased the lateral mobility of AMPARs after LTD induction. To examine the mechanisms underlying this change on lateral diffusion the authors examine the distribution of PSD-95 after LTD induction. They found that after NMDAR, but not P2XR, induced LTD that PSD-95 numbers in the PSD and in nanodomains decreased. The authors then studied the role of PSD-95 phosphorylation by GSK3 beta and autophagy on the observed decreased PSD-95 clusters. Using PSD-95 T19A mutants they found that phosphorylation of this site appeared to be critical for the PSD-95 decrease and also blocked changes in mEPSC amplitude. This mutation also blocked the NMDAR induced change in the mobility of the AMPARs. In contrast, these manipulations did not affect the P2XR induced LTD. This regulation appears to involve autophagy as purification of autophagic vesicles showed that NMDAR, but not P2XR, induced LTD resulted in increased levels of T19 phosphorylated PSD-95 in the vesicles. To examine the role of phosphorylation in these processes they inhibited GSK3beta, the kinase that phosphorylates T19, and showed that it blocked the maintenance of LTD. Inhibitors of autophagosomes similarly inhibited NMDAR induced LTD.

Finally they examined this reorganization of PSD-95 and lateral mobility of AMPARs on short term plasticity, which has been reported to depend on AMPAR lateral mobility, and found that NMDAR, but not P2XR, induced plasticity modifies short term plasticity and increased the paired pulse ratio (PPF) after LTD induction.

The manuscript uses an impressive range of techniques and although some of the concepts have previously been reported, for example the role of autophagy in NMDAR dependent LTD, the results in this manuscript are novel and dissect the mechanisms involved in detail and clearly demonstrate

that NMDAR and P2XR dependent LTP have distinct mechanisms for the maintenance of synaptic depression. Importantly these distinct mechanisms have differential impacts on short term plasticity.

Point-by-point response to reviewers

Nature communication manuscript: NCOMMS-20-35301A-Z

Reviewer's comments have been renumbered for easy reference

Responses to reviewers and additive experiments follow the reviewer comments and all text modifications have been highlighted in yellow in the manuscript.

First, we would like to thank the reviewers for their very helpful comments. We tried to answer all of their questions as extensively as possible. We performed new sets of experiments, including electrophysiology, super-resolution imaging and various new analyses. In parallel, we completed and rewrote partially the manuscript to address the raised issues and improve its readability. We hope that the reviewers will find answers to all their questions and that the manuscript will be suitable for publication. Detailed answers to each specific comments are listed below.

REVIEWER COMMENTS

Reviewer #1 (Remarks to the Author):

This study has the potential to advance our understanding of cellular and molecular mechanisms underlying LTD. In detail, LTD was chemically induced in cultured hippocampal neurons with either NMDA or ATP (in the presence of a blocker of adenosine receptors) to stimulate the NMDAR or P2X receptors, respectively. Both treatments reduced the number of AMPAR in postsynaptic nanodomains and, more generally, dendritic spines, and the amplitude of mostly AMPAR-mediated mEPSCs. NMDA but not ATP increases lateral mobility of GluA2-containing AMPARs. In parallel, PSD-95, the main anchoring protein for AMPAR at postsynaptic sites, also showed a reduction in localization to PSDs and its nanodomains therein following NMDA but not ATP treatment. These results are consistent with earlier findings and expand those by analyzing AMPAR content in nanodomains and AMPAR mobility.

Earlier work showed that LTD requires that PSD-95 is phosphorylated on Thr19 near its N-terminus by GSK3. The authors find that overexpression of PSD-95 with the T19A mutation but not of WT PSD-95 prevented the decrease in AMPARs in nanoclusters, of PSD-95 in PSDs and of mEPSC amplitude induced by NMDA and the increase in AMPAR diffusion. This is again consistent with earlier work.

Perhaps most interesting are data that suggest that WT but not T19A mutant PSD-95 would increase in autophagosomes following NMDA treatment. Application of a GSK3 inhibitor and of a blocker of autophagy prevented the NMDA-induced downregulation of mEPSC amplitude at the 30 min post NMDA treatment time point (although the GSK3 inhibitor did not prevent a temporary reduction in mEPSC at the 10 min time point).

Response: We thank the reviewer for his/her careful read of our manuscript and positive outlook. The point-by-point responses are detailed below.

Major Concern

My main concern is that the most interesting finding that PSD-95 needs to be targeted to autophagosomes for lasting LTD is not sufficiently supported.

1. Subcellular fractionation suggests that PSD-95 is targeted to autophagosomes. The purification of autophagosomes has to be described in detail, not just referenced to earlier work to ensure that there are no differences in the protocols. More importantly, the different fractions have to be more extensively characterized with various subcellular markers than currently provided

We add now a complete description of the autophagosome preparation in the material and methods section (highlighted in yellow in the manuscript) and reported below:

Briefly, mitochondria and peroxisomes are removed with discontinuous Nycodenz gradients. The supernatant was placed on the top of the gradients and was centrifuged at 28,000 rpm for 1h at 4°C. The interface (Aps and endoplasmic reticulum) was isolated and diluted with an equal volume of HB buffer to be loaded on Nycodenz-Percoll gradients in order to remove the small-vesicular and non-membraneous material followed by a 20,000 rpm centrifugation for 30min at 4°C. The interface was collected and diluted with 0.7V of 60% buffered Optiprep and the removal of Percoll silica particles followed by placing 8.5ml of the diluted material in SW40 tubes overlaid with 1.5ml of 30% iodixanol and a top layer of 2.5ml of HB buffer. The material was then centrifuged at 20,000 rpm for 30min at 4°C resulting in the sedimented Percoll particles at the bottom of the tube and the autophagosomes band floated to the iodixanol/HB interface. Autophagosomes were collected for western blot analysis.

We add to the answer to reviewers the requested data on the subcellular markers present into the autophagosome fraction. However, these data are part of another submitted manuscript (currently under positive revision) and available on BioRxiv and referenced in our manuscript (Kallergi et al, <https://doi.org/10.1101/2020.03.12.983965>). We would prefer not to add these data in the current manuscript.

Figure x. Further characterization of the purified AV fraction. Western blot analyses of the fractions obtained during the AV purification procedure with the indicated antibodies, where an equal amount (25µg) of protein from each fraction was loaded. Please note the enrichment in LC3-II in the AV fraction.

2. At the same time colocalization of PSD-95 with autophagosomes has to be demonstrated by immunostaining, ideally with superresolution

We thank the reviewer for this excellent suggestion. We performed dual color d-STORM on PSD95 and LC3, a standard marker for autophagosomes. the results reveal the presence of PSD-95 in autophagosomes both in control and after NMDAR-induced LTD. However, the intensity of PSD-95 after treatment is almost three times the quantity observed in the control condition. These results have been included in the manuscript now as figure 5G and are commented on in the result section (highlighted).

3. For testing whether the LTD they monitor by mEPSC analysis depends on autophagy, the authors use only a single inhibitor (SBI) of autophagy, which could act via side effects rather than by inhibiting autophagy. In fact, there is a tendency for SBI to decrease mEPSC amplitude under basal conditions by itself although it did not reach statistical significance. At least one more inhibitor that acts in a way that is different from SBI is required to be certain enough that autophagy is required for LTD

As requested by the reviewer, we tested the effect of spautin-1 (10μM), another autophagy inhibitor. Spautin-1 targets ubiquitin-specific proteases called USP10 and USP13 (Liu et al., Cell 2011). In normal conditions, these proteases de-ubiquitinate Beclin1 to regulate Vps34, a class III PI3 kinase orchestrating both initiation and maturation of autophagosomes. This inhibition causes an increase in proteasomal degradation of class III PI3 kinase complexes, inhibiting autophagy. In the presence of spautin-1, NMDA-induced LTD is fully abolished. These results have been added in Figure 4S1 and is now referred to in the results.

4. The effect of TDZ8 alone has to be documented in Fig. 5E

Thanks to the reviewer for this comment. We evaluated the effect of TDZD8 alone on a 30-minute timelapse. The results are represented below, and have been added to the supplementary figure 4S1 and is referred to in the manuscript. TDZD8 alone does not trigger any significant effect on mEPSCs.

Modest Concerns

5. It is slightly puzzling that mEPSC frequency was not reduced upon ATP treatment because the observed reduction in mEPSC amplitude should have shifted smaller mEPSC events below the detection threshold given by the electric noise. Perhaps the ATP paradigm effected mostly larger and not smaller mEPSC events. A hint of an analogous potential difference in effects on AMPAR content of larger versus smaller AMPAR nanodomains is perhaps seen in Fig. 1B versus 1H. A more detailed analysis of mEPSC frequency distribution might shed light onto this issue.

We thanks the reviewer for this interesting comment, indeed this point could reveal an important physiological effect.

In our electrophysiology set up, the average noise level is around 2 pA. As shown below, there are very few mEPSCs which are close to 2 pA in amplitude. The majority are comprised between 6 and 12 pA. As LTD triggers, on average, a decrease of 25% of the amplitude, the majority of the mEPSCs should remain detectable after LTD induction.

We tried to clarify this question but none of our new analysis succeeded to obtain a clear explanation of the absence of decrease of miniature events frequency after ATP treatment. Moreover, we did not find a specific decrease of large miniatures compare to small ones, and at the same time, it does not seem that larger nanodomains are more affected by ATP-dependent LTD than the smaller ones.

Distribution of the amplitude before and 10 and 30 minutes after ATP-induced LTD. Dashed line represent the level of electrical noise in our recordings. The number of lost miniatures because of the electrical noise level seems negligible.

Minor Concerns

1. p. 8: the statement "PSD-95 can be immunoprecipitated by LC3" should be better worded, perhaps "PSD-95 co-immunoprecipitates with LC3."

This has been changed.

2. p. 13: "Calcium bound calmodulin interacts with PSD-95 when phosphorylated at T19...." References here are not fully accurate and also not complete (the phosphorylation dependent Calcium / calmodulin binding was shown in Chowdhury et al 2017: EMBO J 37, 122-138, not in the cited paper).

We are sorry for this mistake, this has been changed.

3. Fig 5: the label "T19PSD95" would be clearer as "pT19 PSD95."

This has been changed

Reviewer #2 (Remarks to the Author):

In a primary cell culture model of NMDA or ATP induced depression of synaptic transmission, the authors show that the two forms use different mechanisms to reduce synaptic AMPA receptors. While NMDA reduces PSD-95 and increases the mobility of AMPA receptors, ATP does not affect PSD-95 and AMPA receptor mobility. The author further show that overexpressing PSD-95 T19A impairs the NMDA mediated synaptic depression selectively and it is also impaired by pharmacological blockade of GSK-3 and autophagosome generation. This study is a continuation on a number of studies that aimed to establish that NMDA-dependent long-term synaptic depression is caused by the synaptic removal of PSD-95 as a slot for AMPA receptors. The novel finding here is that after NMDA treatment, AMPA receptor mobility increases and that this prevails throughout the expression of the depressed synaptic state. The authors corroborate this finding by imaging the diffusion of endogenous AMPA receptors with uPaint and alterations in short-term synaptic plasticity.

1. While this finding is novel and interesting, its significance would rise from an agonist induced cell culture model to a potential mechanism of learning and memory, if the authors could show that it also applies to synaptically triggered NMDA receptor-dependent LTD.

Response: We thank the reviewer for his/her careful read of our manuscript and positive feedback. To try to answer to the point concerning the transfer of our conclusion to a more integrated model, we performed electrophysiological experiments on acute brain slices by induction of LTD with the well described low frequency protocol (1 Hz for 10 minutes) and measured its effect on the paired pulse response. As observed with NMDA-induced LTD, we observed an increase in paired pulse ratio, 30 minutes after LFS-induced LTD. This result has now been added to the manuscript as Figure 6I, J and K. Results, discussion and figure legends have been modified to add this new set of data. The new text is highlighted in the manuscript.

LFS-induced LTD protocol

2. Include different proposed mechanisms of PSD-95 removal from PSD for LTD as previously proposed in discussion and how they relate to the new proposed mechanism, including depalmitoylation, ubiquitination, calmodulin N-terminal masking and T19-dependent mobilization.

We added now in the discussion a new proposed mechanism extracted from the literature concerning the ubiquitination of PSD95. The following text has been added: Previous works demonstrated how post-translational modifications of PSD95 can regulate its amount at the PSD and so be important for LTD. As an example, the ubiquitination by the E3 ligase Mdm2 of PSD-95 removes

it from the PSD and rapidly targets it to degradation by the proteasome (Colledge et al. 2003 neuron).

3. Text lacks reference to corresponding figures in multiple cases.

We added substantial references to figures in the result section (highlighted).

4. Fig.1 Scaling of mEPSC sample traces after NMDA or ATP appears smaller. Is the reduction calculated with correctly scaled responses?

We thank the reviewer for this comment. We verified this point but the calculation of the amplitude is correct, the sensation of scale variation comes from the example traces where the electrical background noise can vary.

5. Also add representative traces in the following figures as based on the problem in Fig. 1, the data in the others cannot be judged.

For space reason on the figures which are already quite crowded, we would prefer to not add extra electrophysiological traces but we put below examples of various traces obtained for a couple of experiments performed in the paper.

6. Fig.5S1 Wrong title and incomplete legend.

We are sorry for the mistake, this has been changed.

7. 109 here no pathway-specific modifications are shown

The word “pathway” has been removed

8. If AMPA receptor is your marker for the nanodomain and you have a decrease in AMPA receptor numbers, how does the diameter not change? Does this mean that AMPA receptors are only removed from the center?

This is an excellent remark from the reviewer, we tried technically to tackle this question but this necessitates a resolution better than our current resolution (around 10 nm), which is not reachable with actual imaging techniques. We think that it could be like “porous gruyere cheese”, some AMPAR can exit from the nanodomains creating holes which cannot be distinguished with imaging, while the overall structure of the nanodomain is maintained. This is at least what is revealed by the analysis.

9. Specify that this mechanism applies only to NMDA. For ATP, the mechanism is not further explored.

We added now another reference showing the role of phosphorylation on ATP-dependent LTD.

10. Stein et al 2003 reported this earlier. Include earlier reference.

This reference has been added.

11. If the ratio of T19/total PSD-95 does not change in the AV, why would that be a specific signal to target them there? You would need to compare the T19/total ratio in AV versus synaptosomes for this conclusion.

In fact, the total PSD95 antibody also recognizes the phosphorylated form so we cannot conclude between “an increase of both the phosphorylated and the non-phosphorylated forms” or just “an increase of the phosphorylated one”. We did experiments with synaptosomes but during the preparation it is known that we lose at least partly the soluble post-synaptic proteins which prevent us to properly estimate the amount of phosphorylated PSD95. The only point that we can extract from the data is there is a more than three-fold increase in total PSD95 which is correlated with the three-fold increase of the phosphorylated form.

12. these numbers do not match the experimental data of this study

If the reviewer refers to the number of injected AMPAR and PSD-95 into the model, they do correspond to the experimental data. Regarding AMPAR numbers, we found an experimental number of receptor per spine = 59.8 at the surface, which correspond to the 120 receptor released in

the model to obtain an equilibrium with 60 AMPAR at the surface (~20 trapped in nanodomain) and 60 AMPARs in the intracellular pool (as described in the material and method section).

Regarding PSD-95, the experimental data represented on the figure 3 are the mean of the median per cell because they do not follow a normal distribution, while for modelling we injected into the model the mean of the mean per cell values. Below we report the average of PSD-95 molecule per cluster (per PSD) and nanocluster (NC) which correspond to the 200 PSD95 injected (experimental values 171 +/- 20) and the 52 palmitoylated PSD-95 per nanocluster (see figure 7S1F black curve) (experimental values 54 +/- 5).

This has been specified in the result section (highlighted).

Reviewer #3 (Remarks to the Author):

In this manuscript Compans et al. examine the mechanisms underlying NMDAR vs P2XR dependent LTD using cell biology, biochemistry and electrophysiology. In summary, they show that NMDA receptor and P2XR LTD both involve AMPAR endocytosis and reorganization of AMPARs in nanodomains in the postsynaptic membrane but they provide evidence that NMDAR but not P2XR regulates PSD-95 in the PSD by removal of T-19 phosphorylated PSD-95 by autophagosomes.

This manuscript contains a lot of data that provides evidence for these conclusions First, using super resolution (dSTORM) imaging they show that both NMDAR and P2XR induced LTD results in the number of AMPAR in specific nanodomains with little change in the nanodomain size. This was associated with decreases in mEPSC amplitude as well. In contrast NMDAR but not P2XR induced LTD increased the lateral mobility of AMPARs after LTD induction. To examine the mechanisms underlying this change on lateral diffusion the authors examine the distribution of PSD-95 after LTD induction. They found that after NMDAR, but not P2XR, induced LTD that PSD-95 numbers in the PSD and in nanodomains decreased. The authors then studied the role of PSD-95 phosphorylation by GSK3 beta and autophagy on the observed decreased PSD-95 clusters. Using PSD-95 T19A mutants they found that phosphorylation of this site appeared to be critical for the PSD-95 decrease and also blocked changes in mEPSC amplitude.

This mutation also blocked the NMDAR induced change in the mobility of the AMPARs. In contrast, these manipulations did not affect the P2XR induced LTD. This regulation appears to involve autophagy as purification of autophagic vesicles showed that NMDAR, but not P2XR, induced LTD resulted in increased levels of T19 phosphorylated PSD-95 in the vesicles. To examine the role of phosphorylation in these processes they inhibited GSK3beta, the kinase that phosphorylates T19,

and showed that it blocked the maintenance of LTD. Inhibitors of autophagosomes similarly inhibited NMDAR induced LTD.

Finally they examined this reorganization of PSD-95 and lateral mobility of AMPARs on short term plasticity, which has been reported to depend on AMPAR lateral mobility, and found that NMDAR, but not P2XR, induced plasticity modifies short term plasticity and increased the paired pulse ration (PPF) after LTD induction.

The manuscript uses an impressive range of techniques and although some of the concepts have previously been reported, for example the role of autophagy in NMDAR dependent LTD, the results in this manuscript are novel and dissect the mechanisms involved in detail and clearly demonstrate that NMDAR and P2XR dependent LTP have distinct mechanisms for the maintenance of synaptic depression. Importantly these distinct mechanisms have differential impacts on short term plasticity.

We thank the reviewer for his/her careful reading and positive comments on our manuscript, we have now added a substantial number of new experiments detailed above to improve the overall clarity and quality of the paper and to explain more precisely some aspects of the experiments. All modifications are highlighted in yellow.

Reviewer #1 (Remarks to the Author):

The authors thoroughly addressed the previous concerns. Most reassuring is that they now supplement their evidence for autophagosome targeting of PSD-95 upon induction of LTD with NMDA, which was based on subcellular fractionation and immunoblotting, with superresolution imaging showing increased colocalization of PSD-95 with LC3 as a marker for autophagosomes. They also added evidence that a second inhibitor of autophagosomal protein degradation also inhibits NMDA-induced LTD.

Minor Concerns perhaps for proof reading

Fig 1S1 and S2: abbreviations for water are not always correct like the 2 in H₂O is not always subscripted and O is sometimes given as the number 0.

Fig 4S1: I would spautin-1 not call a “highly potent” inhibitor given that it is used at 10 μ M, a rather high concentration for a pharmacological inhibitor.

Reviewer #2 (Remarks to the Author):

Revision comments:

The authors addressed most of my concerns. However, two issues remain that can be addressed in the text.

1) Classically NMDAR-dependent LTD in the SC-CA1 synapses does not exhibit a change in paired-pulse ratio. The authors need to discuss why they observe an increase in PPR by using similar induction protocols.

2) The evidence that T19 directs to the AV remains vague. If both the total amount of PSD-95 and the T19 phosphorylated increases in the fraction, then there is no specific targeting of the T19 phosphorylated ones. The authors need to adjust their claims here.

Reviewer #1 (Remarks to the Author):

The authors thoroughly addressed the previous concerns. Most reassuring is that they now supplement their evidence for autophagosome targeting of PSD-95 upon induction of LTD with NMDA, which was based on subcellular fractionation and immunoblotting, with superresolution imaging showing increased colocalization of PSD-95 with LC3 as a marker for autophagosomes. They also added evidence that a second inhibitor of autophagosomal protein degradation also inhibits NMDA-induced LTD.

Minor Concerns perhaps for proof reading

Fig 1S1 and S2: abbreviations for water are not always correct like the 2 in H₂O is not always subscripted and O is sometimes given as the number 0.

Fig 4S1: I would spautin-1 not call a “highly potent” inhibitor given that it is used at 10 μM, a rather high concentration for a pharmacological inhibitor.

We thank the reviewer for these remarks, this has been changed.

Reviewer #2 (Remarks to the Author):

Revision comments:

The authors addressed most of my concerns. However, two issues remain that can be addressed in the text.

1) Classically NMDAR-dependent LTD in the SC-CA1 synapses does not exhibit a change in paired-pulse ratio. The authors need to discuss why they observe an increase in PPR by using similar induction protocols.

The reviewer is right there is no dogma on the effect of LTD protocols on paired pulse ratio (PPR). Going through the literature, we found some examples of LTD protocols which trigger an increase of PPR, they are listed below:

Choi et al. 1997, LTD induced by HFS and depolarisation at the striatum

Nosyreva et al. 2005: chemical LTD on hippocampal brain slices (CA1)

Zhang et al. 2006: chemical LTD by NMDA treatment on hippocampal brain slices

Padamsey et al. 2017: LTD induced by HFS at CA1

Andrade-Talavera et al 2016 : LTD induced by STDP at CA3-CA1 synapses

Park et al. 2017: LTD induced by LFS in lateral Habenula

Penasco et al 2019 : LTD induced by LFS at the dentate gyrus

The molecular mechanism of the PPF has been reported as pre or post-synaptic, going through endocannabinoid system or others. Our vision is that PPR integrates a lot of pre and post-synaptic parameters which at the end will define the ability of synapses to answer to frequency input, in this paper we reported the fact that AMPAR mobility is affected by LTD protocols and this parameter tends to favour frequency response.

2) The evidence that T19 directs to the AV remains vague. If both the total amount of PSD-95 and the T19 phosphorylated increases in the fraction, then there is no specific targeting of the T19 phosphorylated ones. The authors need to adjust their claims here.

As requested we changed our sentence in the discussion : “T19 phosphorylated PSD-95 is then targeted to autophagosomes for degradation.” To “PSD-95 is then targeted to autophagosomes for degradation.”